**Subject Category:**
Biology (whole organism)

taxonomy and systematics

conservation management, CT scanning, geometric morphometrics, integrative taxonomy, species delimitation, *Tympanocryptis*

**Author for correspondence:**
Jane Melville
e-mail: jmelv@museum.vic.gov.au

# Integrating phylogeography and high-resolution X-ray CT reveals five new cryptic species and multiple hybrid zones among Australian earless dragons

Jane Melville[1], Kirilee Chaplin[1,2], Christy A. Hipsley[1,2], Stephen D. Sarre[3], Joanna Sumner[1] and Mark Hutchinson[4]

[1]Department of Sciences, Museums Victoria, Carlton, Victoria 3052, Australia
[2]School of BioSciences, University of Melbourne, Parkville, Victoria 3010, Australia
[3]Institute for Applied Ecology, University of Canberra, Canberra, Australian Capital Territory 2601, Australia
[4]South Australia Museum, North Terrace, Adelaide, SA 5000, Australia

  JM, 0000-0002-9994-6423; JS, 0000-0002-0498-6642

Cryptic lineages, comprising species complexes with deep genetic structuring across the landscape but without distinct morphological differences, impose substantial difficulties for systematists and taxonomists in determining true species diversity. Here, we present an integrative approach that combines data from phylogeography and geometric morphometric analyses of three-dimensional cranial models to revisit the uncertain taxonomy of earless dragons from southern and central Australia that at one time or another have been included under the name *Tympanocryptis lineata*. Our approach finds strong support for seven previously described species, and more importantly, five undescribed *Tympanocryptis* taxa for which we provide a taxonomic treatment. We also find evidence of introgression and hybridization in three discrete contact zones between lineages, supported by mitochondrial and nuclear genes, as well as morphological analyses. With a sampling design that includes at least five individuals for each genetic lineage with corresponding X-ray microcomputed tomography scans, we perform comparative evolutionary analyses to show that there is a significant phylogenetic signal in *Tympanocryptis* cranial shape. Our results demonstrate the importance of

using multiple specimens in each genetic lineage, particularly in cases of potential hybridization, and that geometric morphometrics, when used in an integrative framework, is a powerful tool in species delimitation across cryptic lineages. Our results lay the groundwork for future evolutionary studies in this widespread group across multiple environmental types and identify several species of immediate conservation concern with a focus on *T. petersi* sp. nov. We suggest that this species has undergone significant population declines and warrants a full conservation assessment.

## 1. Introduction

Morphologically cryptic lineages, which are species groups with deep genetic structuring across the landscape but that lack distinct morphological differences, present serious challenges for systematists and taxonomists in determining true species diversity [1–3]. Such lineages require robust whole-evidence approaches for identification [4,5], with integrative taxonomy now being a widely used approach [6–8]. However, morphological similarities between genetic lineages continue to impose difficulties for integrative taxonomy, with quantification of external morphological measurements often resulting in little detectable morphological difference (e.g. [2,3]).

High-resolution X-ray microcomputed tomography, or micro-CT, is a promising addition to the range of sources for morphological data in integrative taxonomic research. This method allows non-destructive sampling of museum specimens and clear visualization of the internal skeletal characters that shape external morphology. Micro-CT has been used increasingly in studies of morphological evolution (e.g. [9–13]), but its application to cryptic taxa is still in its early stages [3,14–16]. Geometric morphometric analysis of CT-generated three-dimensional (3D) models has significant potential for species recognition (e.g. [3,17,18]), provided that intraspecific variation can be accounted for. Here, we explore the possibilities of this approach by sampling multiple individuals within genetically discovered clades to incorporate cranial shape diversity as a new source of morphological evidence to partition a complex of Australian earless dragon lizards (Agamidae: *Tympanocryptis*).

This cryptic group of small agamid lizards presents a taxonomic conundrum that necessitates an integrated and high-resolution approach. The species group is polyphyletic but is united by the historical mishap that all were initially described as either a subspecies of *Tympanocryptis lineata* or have been seen as local variations of the nominate subspecies [19]. Two morphological features have been used to group these lizards: (i) a lack of femoral pores and (ii) a dorsal colour pattern consisting of five longitudinal narrow pale stripes that cut across five dark dorsal cross-bands, with a tendency for the light lines to be brighter and better defined as they cross the dark cross-bands. This colour pattern is well developed in populations from the Adelaide region in southern Australia, which is long believed to be the source area of the type specimen of *Tympanocryptis*, *T. lineata* [20]. Hence, an informal treatment of these taxa as members of a *T. lineata* species group or complex, and a tendency to place new taxa of uncertain status as subspecies of *T. lineata* (e.g. [21]) has prevailed.

Two species, *T. houstoni* and *T. pinguicolla*, were found to be morphologically distinct from *T. lineata* [22], with recent work indicating that the disjunct populations assigned to *T. pinguicolla* were sufficiently genetically and morphologically unique to warrant their elevation to species status [19,23]. The recognition of those cryptic species within *T. pinguicolla* combined with the demonstration that the lectotype specimen (ZMB 740) does not come from the published type locality near Gawler, South Australia, but instead from the southeastern highlands of New South Wales, has altered the nomenclatural situation substantially. Specifically, the name *T. lineata* now applies to a narrowly distributed and endangered temperate grassland species from the Australian Capital Territory and adjacent New South Wales, and is not conspecific with those from the Gawler area currently holding the same name [19]. A taxonomic revision of these populations is therefore urgent. In addition, Melville *et al.* [19] found that there were several other clades that were distinct both from the Gawler-area *Tympanocryptis* as well as the temperate grassland species, including the redefined *T. lineata*, requiring further taxonomic assessment.

Here, we apply an integrative taxonomic approach using multi-gene phylogeography, external morphology and geometric morphometric analyses of micro-CT models, to assess the status of these remaining lineages of the *T. lineata* species complex and to define the evolutionary diversity within this group. Our arrangement of specimens based on genetically typed material reveals a series of correlated external morphological differences between genetically defined populations. Our integrative approach with multiple lines of evidence also identifies apparent introgression at or near some contact zones. We regard the available data as consistent with the existence of genetically and

morphologically cohesive species that have undergone limited local hybridization and gene exchange that is insufficient to be part of continuous gene pools. Accordingly, we provide a taxonomic revision, describing them here as new species.

# 2. Methods

## 2.1. Data sources and phylogeography

A list of all specimens used in this study is provided in electronic supplementary material, table S1, encompassing all named and putative species targeted in our study. Based on initial morphological assessment of the 212 specimens included in the study, we determined the following sampling numbers: Species A ($N = 20$); Species B ($N = 10$), Species C ($N = 16$); Species D ($N = 33$); Species E ($N = 13$); *T. centralis* ($N = 22$); *T. houstoni* ($N = 14$); *T. lineata* ($N = 23$); *T. lineata macra* ($N = 31$); *T. mccartneyi* ($N = 3$); *T. osbornei* ($N = 10$); *T. pinguicolla* ($N = 17$). These specimens from Museums Victoria, South Australia Museum, Australian National Wildlife Collection and Australian Museum were used in genetic, external morphological and osteological geometric morphometric analyses.

The mtDNA phylogeography presented in the current paper uses a dataset published in a taxonomic revision of the grassland earless dragons of southeastern Australia (see [19]). In our current study, we provide further details of phylogenetic relationships and phylogeographic structure, which were beyond the taxonomic focus of [19]. This mtDNA dataset comprises a fragment (approx. 1800 bp) of the mtDNA genome that includes the entire protein-coding gene ND2 (NADH dehydrogenase subunit two), and flanking tRNAs, for 290 *Tympanocryptis* samples as well as samples from two other Australian agamid genera (*Pogona* and *Rankinia*) as outgroups [19]. A Bayesian phylogenetic analysis was employed, with model and partitioning scheme determined using the corrected Akaike information criterion (AICc) in PartitionFinder2 on the CIPRES Science Gateway [24,25]. Bayesian analysis was performed using MrBayes [26] on the CIPRES Science Gateway, with two runs of four independent Markov chain Monte Carlo (MCMC) chains (each 50 000 000 generations long, sampled every 1000 generations), under a GTR + I + G model with flat priors. Tracer v. 1.6 [27] was used to check for stationarity and convergence of the chain outputs. The trees were subject to a 25% burn-in in MrBayes, summarized and posterior probabilities obtained. Pairwise mean uncorrected genetic distance for the mtDNA sequences between described and putative species was calculated in Mega 7 [28]. Where available, genetic data from previously sequenced specimens were used, with GenBank numbers listed in electronic supplementary material, table S1.

The electronic edition of this article conforms to the requirements of the amended International Code of Zoological Nomenclature, and hence, the new names contained herein are available under that Code. This published work and the nomenclatural acts it contains have been registered in ZooBank. The LSID for this publication is: urn:lsid:zoobank.org:pub:FF32EF64-FC27-413C-9435-A3586FB1EB26.

## 2.2. External morphology

Fifteen meristic and metric characters previously used in *Tympanocryptis* taxonomy [29,30] and thought to be potentially diagnostic were recorded: subdigital lamellae (SubL); snout–vent length (SVL); tail length (TL); snout width (SnW); head width (HdW); head length (HdL); head depth (HdD); interlimb length (IntL); forelimb length (upper FL1, lower FL2, foot FL3); and hindlimb length (upper HL1, lower HL2, foot HL3, fourth toe 4TL). In addition to these meristic and metric measures, all specimens were qualitatively assessed for variation in scales, patterns and colour that may prove useful to distinguish species. Vouchers were putatively assigned to species based on mtDNA (where available), geographical location and appearance. Electronic callipers were used for all morphological measures to the nearest 0.1 mm and all bilateral counts and measurements were recorded on the left side (where possible). Non-mature individuals and specimens with missing data were removed from all analyses. All external morphological analyses were run in SYSTAT v. 13.2. Prior to analyses, univariate tests of each morphological variable were used to identify outliers and ensure data did not differ significantly from a normal distribution. To standardize for size, a linear regression was performed on SVL against metric measurements, and the residuals were used in further analyses.

A preliminary principal component analysis (PCA) was performed on the above variables, with the first two axes accounting for over 56% of the total variation. The PC1 and PC2 scores for individuals were each subsequently used in two-way analysis of variances (ANOVAs) testing for an effect of sex, species

and their interaction. The post hoc Tukey tests were also used for pairwise group comparisons of all species and sex combinations. There were significant effects of sex and species (but not the interaction of these) for PC1, which was mainly related to tail and limb lengths, and significant effects of sex and species and their interaction for PC2, which was mostly associated with neck width and the presence of pre-anal pores. Data were therefore split, and males and females were analysed separately. Discriminant function analysis (DFA) was subsequently performed on morphological variables for each sex of all study species to identify the measurements that best separated them.

## 2.3. Cranial geometric morphometrics

CT protocol followed Chaplin et al. [3], whereby specimens were scanned in a Phoenix Nanotom M (GE Measurement & Control, MA, USA) for 600 projections at 55 kV, 400 µA and 500 ms, resulting in a final voxel size of 15 µm. Volumetric reconstructions of crania were generated by datos | x-reconstruction software (GE Sensing & Inspection Technologies GmbH, Wunstorf, Germany) and 3D surface models were exported using VGStudio Max 3.0 (Volume Graphics, Heidelberg, Germany). Forty-nine landmarks (see fig. 2 in [3] and in electronic supplementary material, appendix S2 for landmark locations) were placed across the surface of the cranium using Landmark Editor v. 3.6 (Institute of Data Analysis and Visualization, UC Davis, USA).

Three of the scanned specimens (SAMA R37888, SAMA R44712, SAMA R58187) were slightly damaged so that not all landmarks could be reliably placed (one, one and four landmarks on each cranium, respectively); for these, we used the *estimate.missing* function in the R v. 3.1.2 package 'geomorph' [31], with method set to 'Reg' (multivariate regression). The final 3D landmark coordinates were exported to MorphoJ [32], where they were subjected to a generalized Procrustes fit. The resulting Procrustes coordinates, representing the symmetric component of shape variation after standardizing for position, orientation and scale among landmark configurations [33], were used as shape variables in all analyses. Centroid size, calculated as the square root of the sum of squared distances of each landmark to their centroid, was used as a proxy for individual size.

Procrustes ANOVA models were also constructed in geomorph [31], with group differences evaluated using 10 000 random permutations. Procrustes ANOVAs testing for the effects of sex and centroid size were first performed to ensure that there was no bias due to sexual dimorphism or specimen size. We then tested for shape differences among species, followed by pairwise species comparisons of least-squares means using the geomorph function *advanced.procD.lm*. A PCA was conducted on the shape variables, and changes along PC1 and PC2 were assessed to determine which characters contributed the most to morphological variation. Warped cranial surfaces were generated to represent the positive and negative extremes of each PC axis.

Mitochondrial (ND2) sequence data for all scanned specimens (electronic supplementary material, table S1 indicates scanned specimens) were used in a secondary Bayesian analysis to develop a subset of the whole phylogeny based on availability of the CT data. Two partitions were found using the AICc on PartitionFinder2: the first following a GTR + I + G model (for the ND2 coding region in codon position 3) and the second following a GTR + G model (for other sites including flanking tRNAs). Bayesian analyses were performed using MrBayes on the CIPRES Science Gateway, using these two partitions with two runs of four independent MCMC chains (each 50 000 000 generations long, sampled every 1000 generations). Tracer v. 1.6 [27] was used to check for stationarity and convergence of the chain outputs. The trees were subject to a 25% burn-in in MrBayes, summarized and the posterior probabilities obtained. This phylogeny was projected into the PC plots to visualize evolutionary relationships among specimens in relation to their positions in morphospace. The multivariate K-statistic for phylogenetic signal, $K_{mult}$ [34], was used to test the strength of the relationship between genetic divergence and cranial shape based on the above phylogeny and the Procrustes shape coordinates, as well as individual centroid size.

## 2.4. Nuclear DNA sequencing and phylogenetic analysis

In addition to the previously published mtDNA data available for a subset of specimens selected to encompass all mtDNA lineages [19] plus possible hybrid individuals (determined with a combination of morphology and mtDNA), we sequenced a 1226 bp of recombination activating gene-1 (RAG1) exon in the N-terminal domain. Oligonucleotide primer pairs used in PCR amplification and sequencing are detailed in [35,36]. Sequence chromatograms were edited using Geneious v. 10.2.2 (Biomatters Ltd) to produce a single continuous sequence for each specimen.

RAG1 was found to follow the GTR + I + G model of substitution with no partitioning schemes using the corrected Akaike information criterion (AICc) on PartitionFinder2 on the CIPRES Science Gateway [24,25]. Bayesian analysis followed the same protocol as the previous section (see above), except using a GTR + I + G model with flat priors. In addition, a maximum-likelihood analysis was run using RAxML v. 8.2.11 [37] in Geneious v. 10.2.2. For that, a GTR + I + G model was used with 10 000 bootstrap replicates to determine branch support.

## 2.5. Integrative taxonomy

The use of the integrative taxonomic (ITAX) approach, which uses as many lines of evidence as available to delimit species, has proved a powerful tool in taxonomic decision-making in cryptic species complexes [38]. We used an ITAX assessment method previously used in *Tympanocryptis* [29], originally outlined in Miralles & Vences [38], where they specify criteria incorporating both sympatric and not-necessarily sympatric putative species. The assessment method in Melville *et al.* [29] outlines that at least two lines of independent evidence are required to delimit taxa irrespective of sympatry. To delimit species in the current study, we use phylogenetics (mtDNA and nuclear), external morphological data and geometric morphometric data describing cranial shape.

# 3. Results

## 3.1. Mitochondrial phylogeography

To investigate phylogenetic relationships among the *T. lineata* group, we undertook a detailed reassessment of a mtDNA (ND2) phylogenetic analysis [19]. Specifically, a Bayesian inference phylogeographic analysis of 290 *Tympanocryptis* samples and outgroups from Australian agamid genera was performed (figure 1*a*). Here, we present this analysis with expanded details of clades that were not provided in the Melville *et al.* study [19]. The named taxa of the '*T. lineata* group' fall into three major phylogenetic lineages within *Tympanocryptis* (figure 1*a*): (i) *T. l. macra*, from northern Australia, forming a basal clade within *Tympanocryptis* (posterior probability 100%); (ii) *T. lineata*, *T. houstoni*, *T. pinguicolla*, *T. osbornei* and *T. mccartneyi* belonging to a diverse eastern/southern Australian lineage (posterior probability 100%); and (iii) *T. l. centralis*, belonging to the pebble dragon clade, which is widely distributed in stony deserts across the arid zone (posterior probability 100%). However, within the second two of these lineages (figure 1*b,c*), our analysis indicates there is greater phylogenetic diversity than was identified in the recent taxonomic revision [19]. In particular, we identified a further four highly supported clades (posterior probability greater than 98%) within the eastern/southern Australian lineage (figure 1*b,d*), which currently includes *T. lineata*, *T. houstoni*, *T. pinguicolla*, *T. osbornei* and *T. mccartneyi*. This total of nine lineages was supported as four geographical species groups (posterior probability greater than 97%): (i) *T. houstoni*; (ii) Species A and Species B; (iii) Species C and Species D; and (iv) the grassland earless dragons or GEDs (*T. lineata*, *T. pinguicolla*, *T. osbornei* and *T. mccartneyi*). All of the unnamed linages received strong phylogenetic support as being monophyletic, with the mean uncorrected genetic distances between lineages of 6.2–15.5% (table 1).

Although the pebble dragon clade from the arid zone has already had taxonomic work completed recently [30] our additional genetic data, including the first for *T. gigas* (figure 1*a*), provide strong evidence of two highly supported (posterior probability 100%; figure 1*c,d*) geographically delineated clades in central Australia: (i) *T. l. centralis* and (ii) Species E.

## 3.2. External morphology

### 3.2.1. Multivariate analyses

The 12 lineages (7 named, 5 unnamed) putatively assigned to species based on mtDNA, geographical location and appearance could be distinguished from each other in the discriminant function analyses of external morphological measurements for both males (Wilks' $\lambda_{14, 11, 83} = 0.005$, $F_{154,630} = 3.484$, $p < 0.001$) and females (Wilks' $\lambda_{15, 10, 50} = 0.002$, $F_{150,319} = 2.458$, $p < 0.001$). The DFAs correctly classified 77% (75/98) of males and 87% (55/63) of females into the assumed *a priori* species classes. In males, canonical factors 1 and 2 were associated most with neck width and tail length, while in females, canonical factors 1 and 2

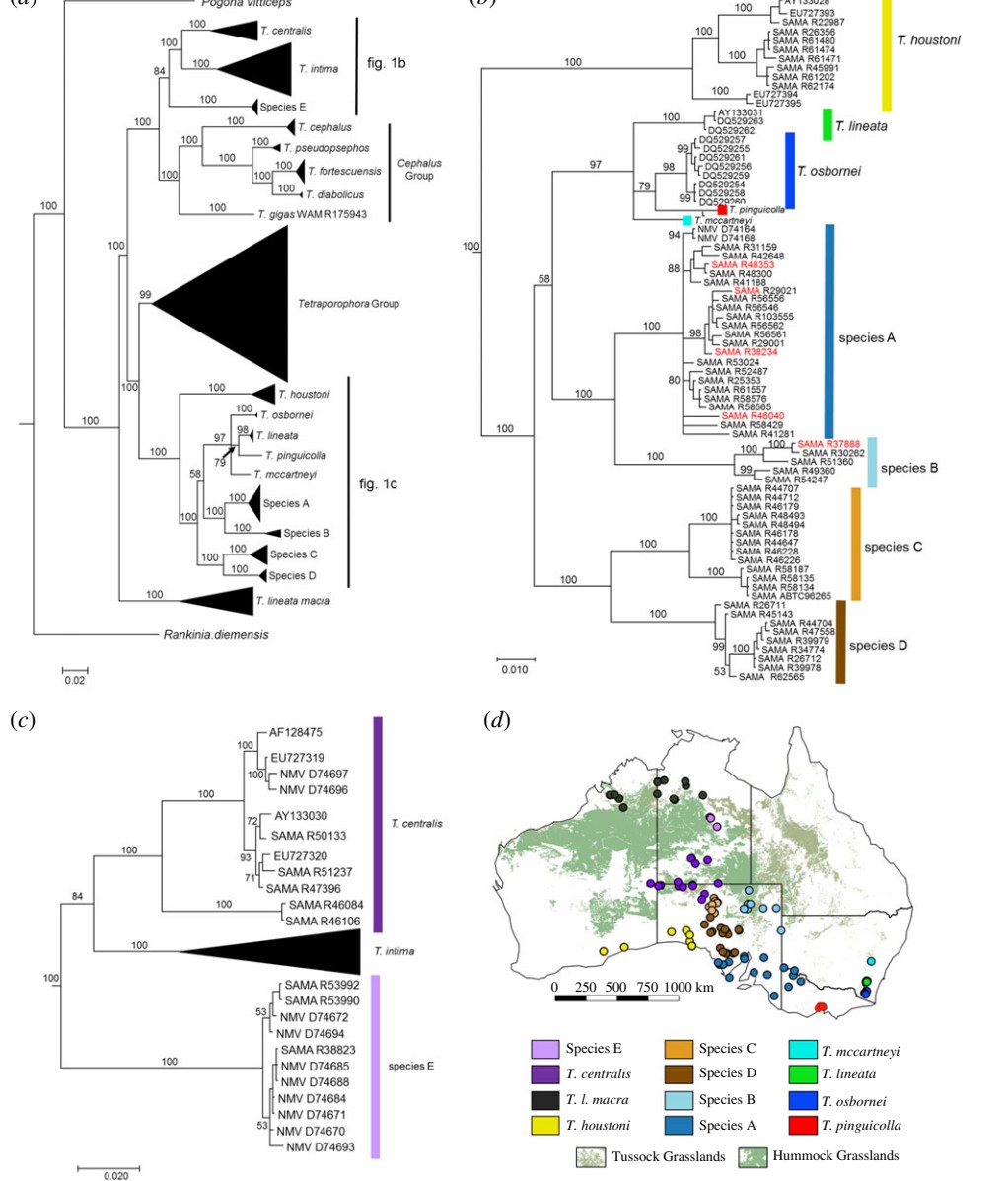

**Figure 1.** Bayesian phylogeny of described and putative *Tympanocryptis* species using approximately 1800 bp of the mtDNA sequence including the gene *ND2* [19]. Posterior probabilities are indicated on nodes, and the scale bar represents the number of nucleotide substitutions per site. Presented are: (*a*) an overall tree with all *Tympanocryptis* species and outgroups included in analyses; (*b*) an expanded subtree of the '*T. lineata*' group with probable hybrid animals highlighted in red; (*c*) an expanded subtree of the '*T. centralis*' group and (*d*) a distribution map of the earless dragon specimens included in the current study, overlain on estimated tussock and hummock grassland habitats [42].

were associated most with neck width and hindlimb length. When the two phylogenetic lineages were analysed separately, the DFAs had higher rates of correct classification for: (i) central pebble species (*T. centralis* and Species E) and (ii) southern species (*T. houstoni*, Species A, B, C and D).

The DFA of the two pebble species was unable to distinguish between the groups based on external morphology for males (Wilks' $\lambda_{14, 1, 16} = 0.220$, $F_{14,3} = 0.760$, $p = 0.692$) or females (Wilks' $\lambda_{7, 1, 7} = 0.008$, $F_{7,1} = 17.373$, $p = 0.183$), even though the DFA correctly classified 94% (17/18) of males (figure 2*a*) and 100% (9/9) of females (figure 2*b*) into the assumed *a priori* species classes. Small sample sizes probably underlie this failure to discriminate between the groups. When considering males only, all *T. centralis* were classified correctly but one individual of Species E (NMV D74685) was misclassified as a *T. centralis*. For both males and females, canonical factor 1 is not shown as it accounted for all of the variance (figure 2). Male *T. centralis* have shorter hindlimbs and longer forelimbs than Species E, while female *T. centralis* had longer tails and shorter heads and hindlimbs than Species E.

**Table 1.** Pairwise mean uncorrected genetic distance between described and putative *Tympanocryptis* species based on approximately 1800 bp of the mtDNA sequence including the gene *ND2* (NADH dehydrogenase subunit two) and flanking genes encoding tRNA$^{Trp}$, tRNA$^{Ala}$, tRNA$^{Asn}$, tRNA$^{Cys}$, tRNA$^{Tyr}$. Species groups correspond to those on the mtDNA phylogenetic tree (figure 1*a*).

| | | 1 | 2 | 3 | 4 | 5 | 6 | 7 | 8 | 9 | 10 | 11 | 12 | 13 |
|---|---|---|---|---|---|---|---|---|---|---|---|---|---|---|
| 1 | *T. tetraporaphora* group | | | | | | | | | | | | | |
| 2 | *T. pinguicolla* | 0.120 | | | | | | | | | | | | |
| 3 | *T. lineata* | 0.126 | 0.054 | | | | | | | | | | | |
| 4 | *T. osbornei* | 0.124 | 0.048 | 0.036 | | | | | | | | | | |
| 5 | Species B | 0.135 | 0.087 | 0.083 | 0.081 | | | | | | | | | |
| 6 | Species A | 0.132 | 0.069 | 0.082 | 0.074 | 0.066 | | | | | | | | |
| 7 | Species C | 0.122 | 0.082 | 0.084 | 0.081 | 0.098 | 0.078 | | | | | | | |
| 8 | Species D | 0.126 | 0.082 | 0.085 | 0.085 | 0.099 | 0.088 | 0.062 | | | | | | |
| 9 | *T. houstoni* | 0.138 | 0.095 | 0.104 | 0.104 | 0.126 | 0.111 | 0.111 | 0.117 | | | | | |
| 10 | *T. intima* | 0.140 | 0.148 | 0.154 | 0.144 | 0.153 | 0.152 | 0.147 | 0.142 | 0.157 | | | | |
| 11 | *T. centralis* | 0.122 | 0.092 | 0.116 | 0.114 | 0.127 | 0.120 | 0.118 | 0.114 | 0.124 | 0.113 | | | |
| 12 | Species E | 0.134 | 0.136 | 0.136 | 0.129 | 0.138 | 0.132 | 0.135 | 0.134 | 0.141 | 0.128 | 0.102 | | |
| 13 | *T. cephalus* group | 0.134 | 0.146 | 0.137 | 0.134 | 0.145 | 0.145 | 0.139 | 0.140 | 0.145 | 0.138 | 0.111 | 0.126 | |
| 14 | *T. lineata macra* | 0.140 | 0.147 | 0.140 | 0.136 | 0.144 | 0.144 | 0.140 | 0.137 | 0.147 | 0.153 | 0.124 | 0.141 | 0.145 |

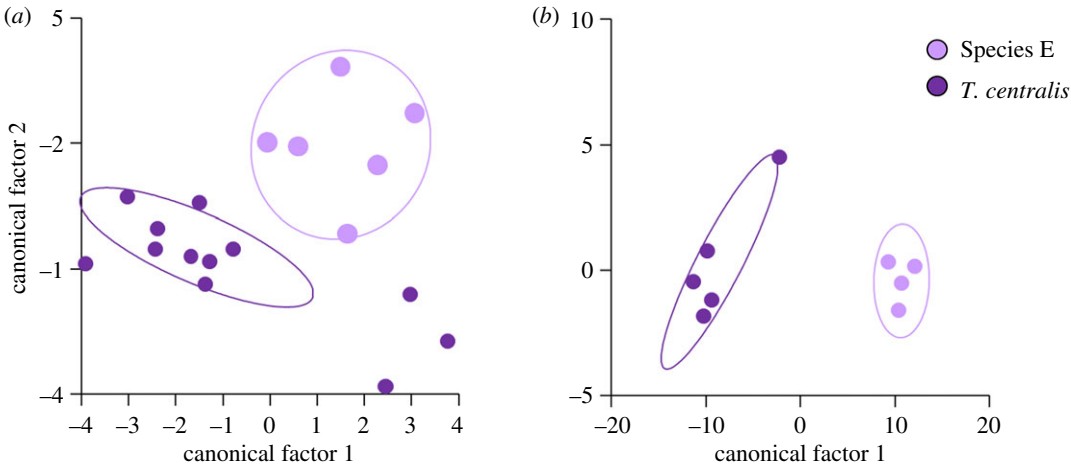

**Figure 2.** Discriminant functional analyses of (*a*) males and (*b*) females of the '*T. centralis*' group, using 15 meristic and metric external morphology measurements: SubL; SVL; TL; SnW; HdW; HdL; HdD; IntL; forelimb length (upper FL1, lower FL2, foot FL3); and hindlimb length (upper HL1, lower HL2, foot HL3, fourth toe 4TL).

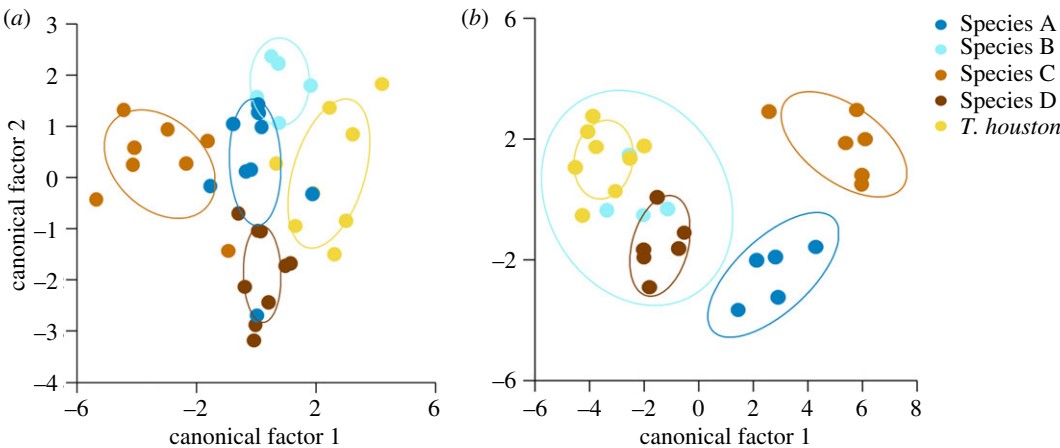

**Figure 3.** Discriminant functional analyses of (*a*) males and (*b*) females of the '*T. lineata*' group, using 15 meristic and metric external morphology measurements: SubL; SVL; TL; SnW; HdW; HdL; HdD; IntL; forelimb length (upper FL1, lower FL2, foot FL3); and hindlimb length (upper HL1, lower HL2, foot HL3, fourth toe 4TL).

The DFA of the five Southern species significantly distinguished groups based on external morphology for males (Wilks' $\lambda_{13, 4, 36} = 0.049$, $F_{52,95} = 2.159$, $p = 0.001$) and females (Wilks' $\lambda_{15, 4, 24} = 0.005$, $F_{60,40} = 1.908$, $p = 0.016$), correctly classifying 83% (34/41) of males (figure 3*a*) and 97% (28/29) of females (figure 3*b*) into the assumed *a priori* species classes. In males, the DFA correctly assigned 100% (5/5) of Species B, 75% (6/8) of Species C (SAMA R5640 as Species F), 66% (6/9) of Species D (NMV D5424 as Species A), 73% (5/7) of *T. houstoni* (SAMA R26356 and R61474 as *T. lineata*) and 66% (6/9) of Species A (NMV D69969 as Species B, SAMA R65809 as Species D and SAMA R41188 as *T. houstoni*). For males, canonical factors 1 (63.3% of variance) and 2 (25.6% of variance), when corrected for within-group variance, were associated most with hindlimb length and tail length. In females, the DFA correctly assigned 100% (29/29) of all species, except Species D (83%), where it incorrectly classified one individual as a Species B (SAMA R67456 from the Gawler Ranges region in South Australia). For females, canonical factors 1 (74.6% of variance) and 2 (17.5% of variance), when corrected for within-group variance, were associated most with hindlimb length and neck width.

### 3.2.2. Qualitative characters

Several scalation features were frequently diagnostically useful within this species assemblage. These features include whether the small dorsal scales are keeled and imbricate or smooth and almost

(a)

Species A
Hambidge CP
(SAMA R25354)

Species B
Kalamurinna
(SAMA R54258)

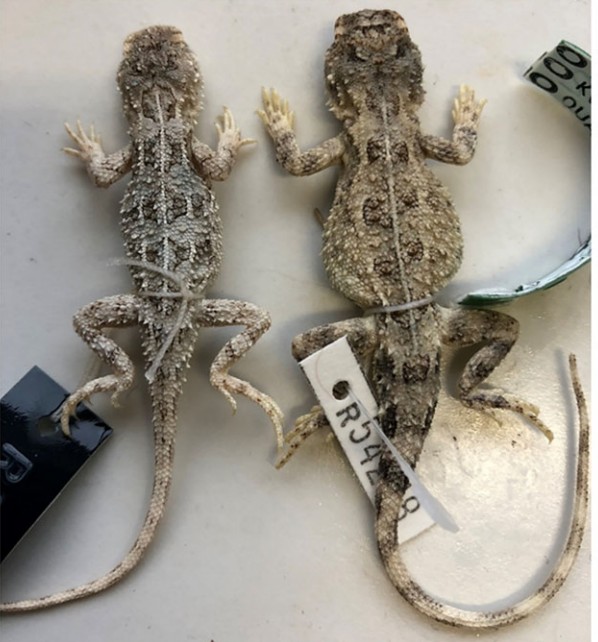

(b)

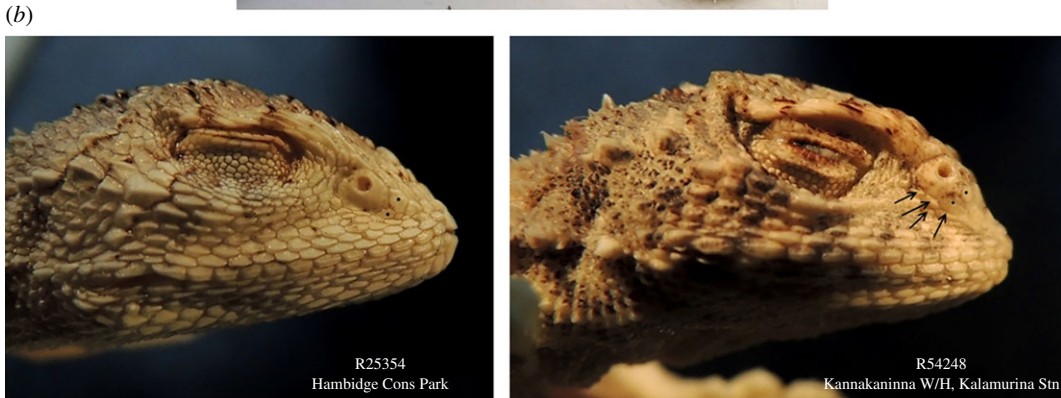

R25354
Hambidge Cons Park

R54248
Kannakaninna W/H, Kalamurina Stn

**Figure 4.** Dorsal pattern and colour can be very similar in Species A and Species B (a), but lateral views of heads (b) show that Species B has a more convex snout profile, and heterogeneous scalation below the nasal scale (arrows).

juxtaposed and whether the scalation on the dorsal surface of the head is strongly keeled or smooth to weakly keeled. The position of the nasal scale with respect to the canthus rostralis, the size of the scales that border its ventral margin, the degree of differentiation of these subnasal scales and the smaller rows of scales that cover the snout between the nasal and the supralabials (figure 4) are a second set of useful variations. The supralabial scale rows themselves may be margined dorsally by two scale rows that are continuous around the front of the snout, and crossing above the rostral scale to separate it from the canthus rostralis. This causes the snout shape in profile to be altered from smoothly tapering to a point to a more convex shape with a distinctly protruding rostral and labial region.

Pattern variation is variably sexually dimorphic, with females generally having a duller and more weakly lined version of the colour pattern seen in males of the same population. Other morphological features expressed in most species to some degree include the frequent presence of a lateral skin fold along the neck, and a colour pattern including a pale transverse bar that crosses the top of the head over the eyes. All lack femoral pores and have just two precloacal (pre-anal) pores.

## 3.3. Cranial shape variation

There was no evidence of sexual dimorphism in cranial shape among the scanned specimens ($R^2 = 0.032$, $p = 0.1557$), and a significant but weak effect of log-transformed centroid size ($R^2 = 0.0746$, $p = 0.0012$). By

contrast, there was a highly significant and strong effect of species ($R^2 = 0.5048$, $p < 0.0001$), with pairwise tests indicating cranial shape differences between 16 of the 45 pairwise comparisons (table 2). Of those, the largest shape differences were between *T. lineata* and Species B, C and D; *T. pinguicolla* and Species C and D; and *T. l. macra* and Species A, B, C, D and *T. houstoni*. The remaining significant pairwise shape differences were between *T. houstoni* and Species B and C, and Species E compared with Species A, B, C and *T. l. macra*.

Taxonomic groupings in morphospace appear to reflect the above pairwise results (detailed further in Basis of taxonomic decisions), with the first two PC axes explaining 36% of the total shape variance (figure 5). PC1 was associated with changes in cranial width and length, influenced by breadth of the temporal region (postorbital and jugal bones), and anteroposterior extent of the snout (premaxilla). Individuals with positive PC1 scores tended to have blunt, snub-nosed faces with short occipital regions and a more bulbous skull (Species A–D), while those with negative PC1 scores were more tapered and elongate (*T. l. macra*, *T. lineata*, *T. pinguicolla*, Species E). Changes along PC2 were more subtle, related to the width and height of the back of the skull along the parietal table and occipital region. There was a non-significant phylogenetic signal in log-transformed centroid size ($K_{mult} = 0.0237$, $p = 0.4109$), and a highly significant but weak phylogenetic signal in cranial shape ($K_{mult} = 0.0392$, $p < 0.0001$).

## 3.4. Nuclear DNA phylogenetics

Seventy-seven new and previously published sequences were analysed for the RAG1 protein-coding gene to further explore phylogenetic and potential hybridization between putative species (electronic supplementary material, table S1 details specimens sequenced). The alignment comprised 1226 characters: 151 characters were variable and 89 characters were parsimony informative. The maximum-likelihood tree (figure 6; ln-likelihood −3391.04) recovered each of the putative species; however, bootstrap support for the monophyly of Species A and C was 27 and 31%, respectively. The monophyly of Species B, D and E was well supported, with bootstrap values of 95%, 70% and 100%, respectively. Of the already named species, *T. houstoni* was not supported as a monophyletic lineage, probably as a result of incomplete lineage sorting in the nuclear gene.

The phylogenetic relationships between lineages differed between the mtDNA and nuclear phylogeny. Unlike the mtDNA phylogeny, *T. l. macra* was resolved as a member of the pebble dragon clade in the RAG1 phylogeny, along with the *T. cephalus* clade, *T. intima* and Species E. The remaining lineages of *Tympanocryptis* formed a second lineage, within which were two sister lineages: (i) the *T. tetraporaphora* clade and (ii) the remaining *T. lineata*-group samples. This second clade of *T. lineata*-group samples comprises a single basal *T. houstoni* sample and the two sister lineages: (i) other *T. houstoni* samples, the GED clade and Species D and (ii) Species A, B and C.

We included four additional samples in this RAG1 analysis that had been identified as potential hybrids using mtDNA and morphological assessment. Three of these four samples occurred in a different place in the RAG1 tree compared with the mtDNA tree. SAMA R37888 was part of Species B in the mtDNA tree but morphologically identifiable as Species D and in the RAG1 tree, it was resolved as a member of the clade including Species D (figure 6). Similarly, SAMA R38234 and SAMA R29021 were both mtDNA Species A but identified as belonging to the Species D clade both morphologically and in the RAG1 phylogeny. By contrast, the fourth potential hybrid included, SAMA R48353, was identified morphologically as Species B but phylogenetically as Species A in both the mtDNA and RAG1 phylogenies. Thus, in the majority of cases, the RAG1 phylogeny is congruent with external morphology, indicating that the mismatched mtDNA may be the result of interbreeding at contact zones between three of the lineages. The mismatch between the molecular and morphological data in the fourth case can also be explained by interbreeding, but including admixture of nuclear as well as mitochondrial DNA.

## 3.5. Basis of taxonomic decisions

Our study, which incorporated mtDNA, nuclear DNA, external morphology, qualitative scalation and geometric morphometric analyses of 3D cranial shape, shows that the *T. lineata* group is paraphyletic and contains far greater diversity than is presently recognized. Summarized in table 3 are the lines of evidence supporting delimitation of lineages reviewed in the taxonomic section: Species A, B, C, D and E; *T. centralis*; and *T. houstoni*. For the morphological delimitation, including external morphology, qualitative scale morphology and 3D cranial shape, we provide a conservative assessment—only lineages with significant differences in both males and females are scored as distinct. For some of the

**Table 2.** Pairwise tests of cranial shape variation from geometric morphometric analysis of described and putative *Tympanocryptis* species, with least-squares mean distances above the diagonal and *p*-values below. Values in italics indicate significant shape differences.

| | | 1 | 2 | 3 | 4 | 5 | 6 | 7 | 8 | 9 | 10 |
|---|---|---|---|---|---|---|---|---|---|---|---|
| 1 | Species B | — | 0.0364 | 0.0405 | 0.0680 | *0.0516* | *0.0399* | *0.0715* | *0.0818* | *0.0757* | *0.0477* |
| 2 | Species C | 0.2561 | — | 0.0389 | 0.0571 | *0.0498* | *0.0398* | *0.0727* | *0.0905* | *0.0862* | *0.0469* |
| 3 | Species D | 0.6648 | 0.6184 | — | 0.0622 | *0.0492* | *0.0460* | *0.0651* | *0.0860* | *0.0857* | *0.0479* |
| 4 | T. centralis | 0.1706 | 0.3752 | 0.4655 | — | *0.0633* | *0.0595* | *0.0649* | *0.0868* | *0.0869* | *0.0539* |
| 5 | T. houstoni | *0.0182* | *0.0065* | 0.1955 | 0.2331 | — | *0.0305* | *0.0515* | *0.0731* | *0.0603* | *0.0415* |
| 6 | Species A | 0.1347 | 0.0504 | 0.2645 | 0.302 | 0.2084 | — | *0.0586* | *0.0709* | *0.0654* | *0.0427* |
| 7 | T. lineata macra | *<0.0001* | *<0.0001* | *0.0061* | 0.1662 | *0.0017* | *0.0002* | — | *0.0651* | *0.0714* | *0.0473* |
| 8 | T. lineata | *0.0276* | *0.0016* | *0.0443* | 0.1553 | 0.083 | 0.0903 | 0.1639 | — | *0.0651* | *0.0718* |
| 9 | T. pinguicolla | 0.0694 | *0.0059* | *0.0489* | 0.1572 | 0.3066 | 0.1822 | 0.0714 | 0.6757 | — | *0.0691* |
| 10 | Species E | *0.0256* | *0.0090* | 0.1993 | 0.4966 | 0.0545 | *0.0262* | *0.0021* | 0.0825 | 0.1181 | — |

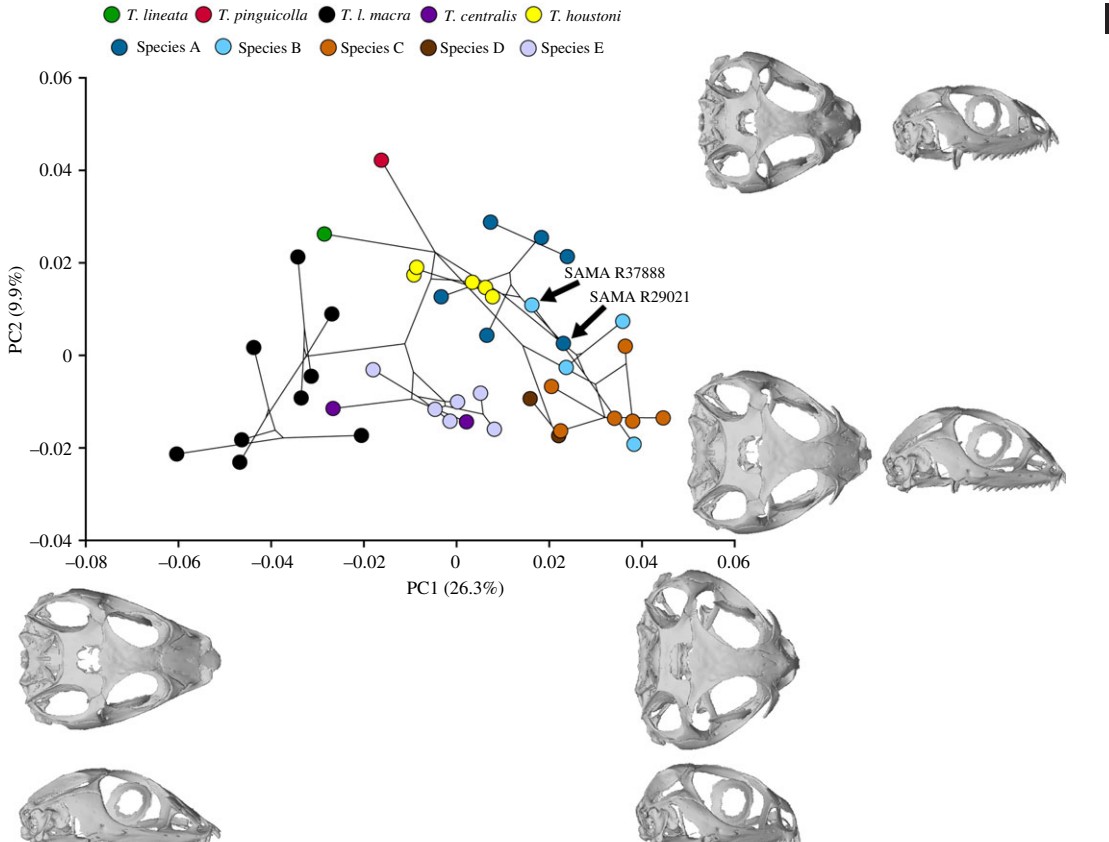

**Figure 5.** *Tympanocryptis* cranial shape morphospace based on PCA of the three-dimensional landmark coordinates (see electronic supplementary material, figure S9 in appendix S2 for the placement of landmarks), with the corrected maximum-likelihood phylogeny projected onto the specimen scores. Warped cranial surfaces representing the extreme shapes at the ends of each axis are shown using the specimen closest to the estimated mean shape for the landmarked dataset (Species E: NMVD74672).

lineages, such as Species B and D, there were significant differences in external morphology between males but not females. In such cases, we did not count this as evidence for delimitation. All lineages have at least one morphological line of evidence supporting it as being a distinct lineage. Similarly, all but one lineage (*T. houstoni*) were monophyletic in both the mtDNA and nuclear DNA phylogenetic analysis. Thus, using an integrative taxonomic assessment criteria (ITAX) previously used in *Tympanocryptis* [3,29], we identified the five existing named taxa and eight additional unnamed species (figure 7).

This assessment included samples which indicated mtDNA introgression between three lineages in areas of sympatry (figure 8). Specifically, sister mtDNA lineages Species A and Species B, which have a region of sympatry in eastern South Australia and western New South Wales, showed evidence of mtDNA introgression (figure 7), supported by RAG1 nuclear evidence. These two individuals (SAMA R46040 and R48353) were morphologically identifiable as Species B, indicating that there is a zone of gene exchange in sympatry between these two lineages. In addition, there is a region of sympatry between Species A and Species D in the Gawler Ranges and the northern Eyre Peninsula, South Australia. In this region, one animal, identified as Species A in the mtDNA phylogeny (SAMA R29021) but associated with Species D in the RAG1 phylogeny, was morphologically identifiable as Species D. This animal also fell outside of the Species A cluster in cranial morphospace (figure 5). In summary, our data provide evidence that Species D and Species A are not mtDNA sister taxa and are readily identifiable by external morphology, but have a broad area of distributional overlap where some gene exchange occurs. Finally, we included one sample that was in the Species B mtDNA lineage from the Olympic Dam area of South Australia but was clearly morphologically identifiable as Species D (SAMA R37888), supported by the nuclear data, and was distant to its congeners in cranial shape (figure 5). There is currently a distributional gap between Species D and Species B, indicating that this instance of mtDNA introgression may represent a past dispersal event of Species B from the Strzelecki Desert in northeastern South Australia to the Olympic Dam area.

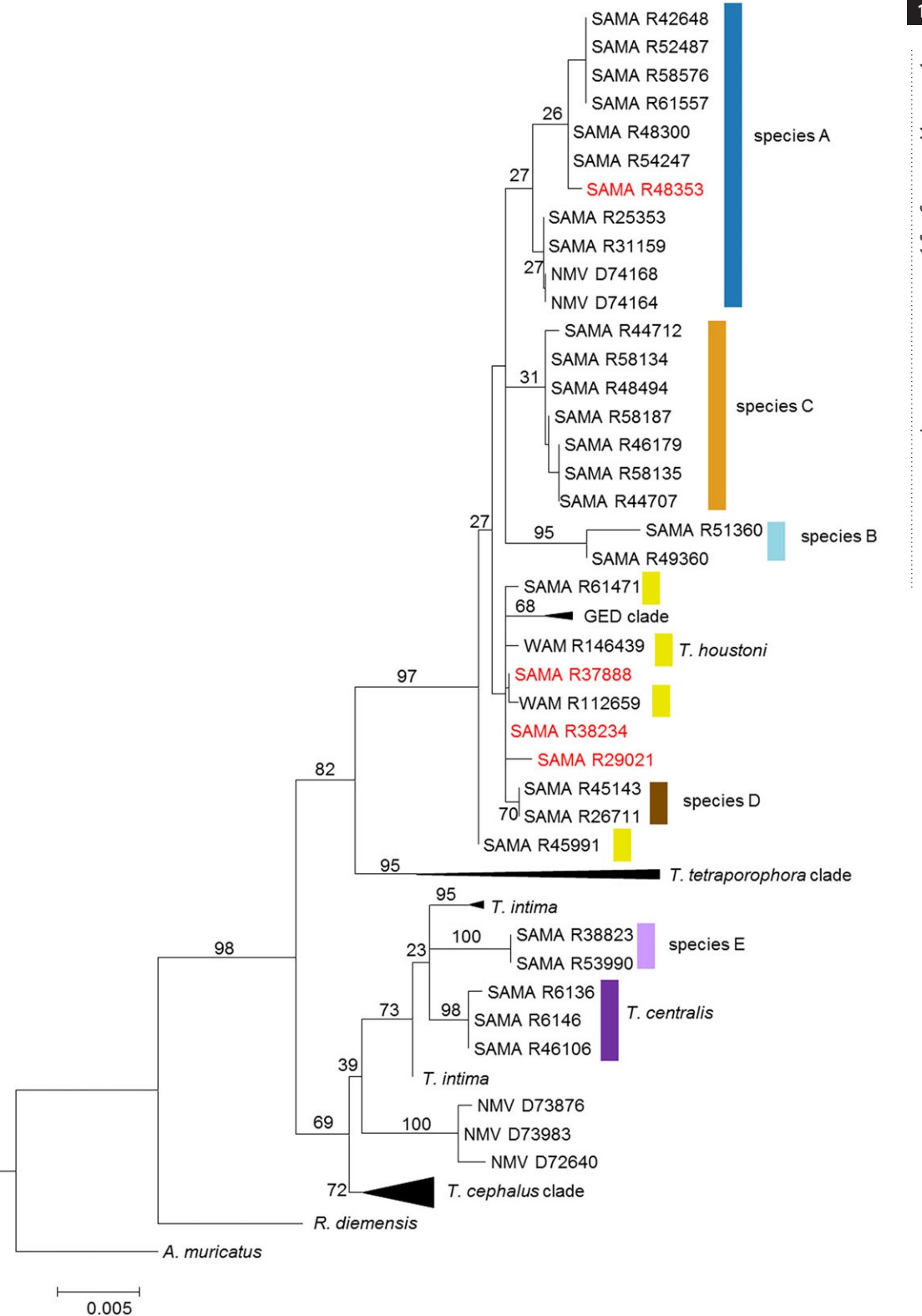

**Figure 6.** Maximum-likelihood phylogeny of described and putative *Tympanocryptis* species using approximately 1200 bp of recombination activating gene-1 (RAG1) exon in the N-terminal domain. Bootstrap branch support is indicated on nodes, and the scale bar represents the number of nucleotide substitutions per site. Probable hybrid animals highlighted in red.

## 3.6. Taxonomic revision

All the species described belong to the genus *Tympanocryptis*, small agamids (max SVL approx. 60 mm) with a relatively short, deep head which lacks a tympanic membrane, the area of the ear being covered by

**Table 3.** Summary of integrative taxonomic assessment. Pairwise summary of morphological delimitation evidence between putative species (Species A, B, C, D, E), *T. centralis* and *T. houstoni*. DFA of external morphology (D), qualitative assessment of scalation summaries in table 5 (S), geometric morphometrics of cranial shape (C). Also provided, whether mtDNA and nuclear DNA support the monophyly of the lineage, indicated with a '✓'. This assessment is also represented on the mtDNA phylogeny in figure 7.

| | species A | species B | species C | species D | species E | *T. centralis* | mtDNA | nuclear DNA |
|---|---|---|---|---|---|---|---|---|
| 1 | Species A | — | | | | | | | ✓ |
| 2 | Species B | D, S | — | | | | | ✓ | ✓ |
| 3 | Species C | D, S | D, S | — | | | | ✓ | ✓ |
| 4 | Species D | D, S | S | D, S | — | | | ✓ | ✓ |
| 5 | Species E | C, D, S | C, D | D, C | | — | | ✓ | ✓ |
| 6 | *T. centralis* | D, S | D, S | D, S | D, S | D | — | ✓ | ✓ |
| 7 | *T. houstoni* | D | C, S | C, D, S | D, S | D, S | D, S | | — |

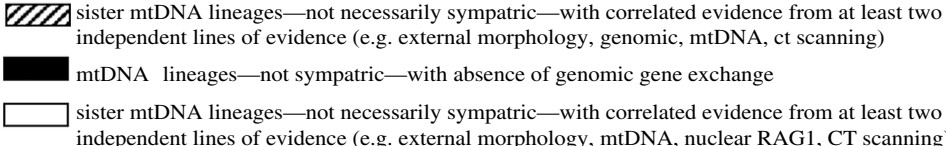

**Figure 7.** Results from the mtDNA phylogenetic analysis with integrative taxonomic (ITAX) assessment of species delimitation. Species are denoted by a coloured block, red and yellow lines represent evidence of introgression between lineages and patterned rectangles on branches denote the evidence for species delimitation.

a continuation of the temporal scalation, and the stapes short, stout and attached to the internal surface of the temporal epidermis. All have a depressed body shape with heterogeneous body scalation due to numerous enlarged spinous scales scattered across the dorsum but not forming continuous rows. Scales on the ventral surface are imbricate and smooth to lightly keeled. Two precloacal pores are

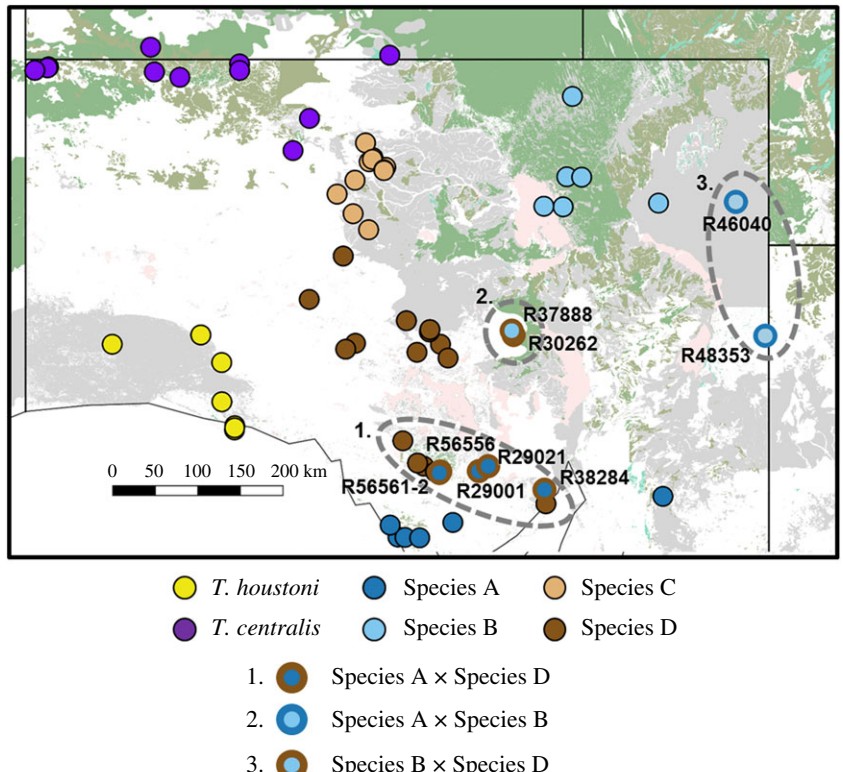

**Figure 8.** Map of South Australia showing the distribution of putative and described *Tympanocryptis* species, with probable zones of hybridization indicated by dashed-line ellipses. Registration numbers of museum specimens that were identified as probable hybrid animals are provided, with a symbol indicating parental species.

present, but femoral pores are absent. Summary morphological characters (table 4) and a guide to distinguishing species based on morphological characters (table 5) are provided. Museum abbreviations used below are ZMB: Museum für Naturkunde, Berlin; NMV: National Museum of Victoria; SAMA: South Australian Museum, Adelaide.

Order Squamata Oppel, 1811
Family Agamidae Spix, 1825
Genus *Tympanocryptis* Peters, 1863

*Tympanocryptis petersi* **sp. nov.**
ZooBank urn:lsid:zoobank.org:act:ECCBCBD1-2F78-4320-9C62-6F02B1DEA242
Lined Earless Dragon
[Referred to as Species A in the Results section]
(figures 9*a* and 11*a*)

**Holotype.** ZMB 54549 (formerly 4714, part), sent from 'Buchfelde', near Gawler, South Australia [approximately 34°35′, 138°31′ S], subadult male. Collector Richard Schomburgk, 1863. Specimen from the syntype series of *T. lineata* [20].

**Paratypes.** ZMB 7414, same data as holotype, adult female. NMV D53031, Big Desert, 2.75 km NNW of Chinaman Well, Victoria, 35°51′24″ S, 141°39′10″ S, adult female; D57507, 14 km W of Hattah, Victoria, 34°46′ S, 142°06′ E, immature male; D59670, 14 km W of Hattah, Victoria, 34°46′ S, 142°06′ E, adult male; D74164, Lake Tyrell, Victoria, 35°20′ S, 142°50′ E, adult female; D74168, Lake Tyrell, Victoria, 35°20′ S, 142°50′ E, adult male; SAMA R31159, ruins at Thompsons Beach Road, South Australia, 34°29′ S, 138°19′ E, adult male; SAMA R42648, Ngaut Ngaut Conservation Park, South Australia, 34°42′ S, 139°37′ E, subadult male; R53024, Deep Swamp, 5.1 km WSW of Alaman H/S, South Australia, 36°44′35″ S, 140°16′25″ E, adult male; R54592, 8 km N of Blanchetown, South Australia, 34°15″ S, 139°36″ E, adult female; R58429, 6.6 km ENE Tungketta Hill, Eyre Peninsula, South Australia, 33°43′16″ S, 135°7′37″ E, adult female; R58576, 5.1 km SSE Horne Lookout, Eyre Peninsula, South Australia, 33°43′41″ S, 135°21′49″ E, adult male; R61557 4.6 km NE Walkers Rock, Eyre Peninsula, South Australia, 33°31′23″ S, 134°53′14″ E, adult male.

**Table 4.** Morphological character measurements of *Tympanocryptis* species described in the current study. Mean ± s.e. provided for measurements, range provided for counts. SubL, subdigital lamellae; SVL, snout–vent length; TL, tail length; SnW, snout width; HdW, head width; HdL, head length; HdD, head depth; IntL, interlimb length; FL, forelimb length; HL, hindlimb length.

| species | N | SubL | SVL | TL | SnW | HdW | HdL | HdD | IntL | FL | HL |
|---|---|---|---|---|---|---|---|---|---|---|---|
| *T. petersi* (Species A) | 15 | 17–22 | 50.8 ± 1.89 | 76.8 ± 2.79 | 5.4 ± 0.19 | 11.9 ± 0.35 | 18.2 ± 0.53 | 8.2 ± 0.16 | 21.9 ± 1.29 | 24.0 ± 1.10 | 37.7 ± 1.35 |
| holotype ZMB 54549 | — | 19 | 50 | 75 | 5.2 | 11.8 | 18.2 | 8.0 | 21.1 | 23.8 | 37.5 |
| *T. argillosa* (Species B) | 8 | 19–23 | 43 ± 3.5 | 64 ± 5.8 | 5.5 ± 0.27 | 11.1 ± 0.52 | 15.9 ± 1.02 | 7.7 ± 0.40 | 18.4 ± 1.72 | 20.8 ± 1.33 | 32.8 ± 2.24 |
| holotype SAMA R49360 | — | 18 | 49 | 68 | 5 | 11.2 | 15.8 | 8.1 | 19.9 | 21.8 | 36.3 |
| *T. fictilis* (Species C) | 16 | 16–23 | 43.6 ± 1.73 | 57 ± 2.96 | 4.7 ± 0.17 | 10.8 ± 0.40 | 15.9 ± 0.63 | 6.8 ± 0.26 | 18.9 ± 0.91 | 22.4 ± 0.75 | 34.2 ± 1.40 |
| holotype SAM R46179 | — | 19 | 46 | 69 | 4.8 | 11.1 | 18.4 | 7.3 | 17.2 | 22.8 | 35.52 |
| *T. tolleyi* (Species D) | 18 | 17–22 | 45.2 ± 0.83 | 68.1 ± 1.70 | 5.1 ± 0.08 | 11.1 ± 0.15 | 16.4 ± 0.25 | 7.35 ± 0.14 | 19.8 ± 0.51 | 21.4 ± 0.31 | 34.3 ± 0.56 |
| holotype SAMA R69147 | — | 19 | 43 | 74 | 4.5 | 10.7 | 14.1 | 6.7 | 18.7 | 19.5 | 31.1 |
| *T. rustica* (Species E) | 10 | 17–21 | 46.9 ± 1.91 | 75.3 ± 2.89 | 4.9 ± 0.17 | 11.5 ± 0.35 | 18.0 ± 0.65 | 7.7 ± 0.28 | 19.9 ± 0.86 | 23.6 ± 0.77 | 37.1 ± 0.89 |
| holotype NMV D74671 | — | 19 | 48 | 78 | 4.9 | 11.7 | 19.3 | 7.7 | 19.3 | 24.8 | 39.8 |

**Table 5.** Summary of characters useful for distinguishing among species of the 'T. lineata' complex. Less common character states are given in italics.

| character | petersi | houstoni | argillosa | tolleyi | fictilis | pinguicolla | lineata | osbornei | macartneyi | centralis | rustica | macra |
|---|---|---|---|---|---|---|---|---|---|---|---|---|
| neck fold continuous to gular fold | yes | yes | yes | yes | no | yes | yes | yes | yes | no | weak | no |
| nasal scale position wrt canthal ridge | below | below | on | on | on | below | below | below | below | on | on | below |
| nasal scale bordered below by enlarged scales | no | no | yes | yes | yes | no | no | no | no | yes | yes | yes |
| dorsal snout scales keeled | yes | yes | yes | yes | no | yes | yes | yes | yes | yes | yes | yes |
| dorsal body scales strongly keeled, overlapping | yes | yes | no | weak or no | no | yes | yes | yes | yes | yes | yes | yes |
| enlarged dorsal spinous scales longitudinally aligned | no | no | no | no | no | no | no | no | no | yes | yes | no |
| spinose scales on thighs | yes | yes | yes | yes | yes | yes | no | no | yes | yes | yes | yes |
| ventral body scales | weakly keeled | smooth | smooth | smooth | smooth | smooth | smooth | smooth | weakly keeled | weakly keeled | weakly keeled | strongly keeled |
| throat scales | smooth | smooth | smooth | keeled | smooth | smooth | smooth | smooth | keeled | keeled | smooth | keeled |
| lateral skin fold from axilla to groin | no | no | no | no | no | yes | yes | yes | yes | no | no | no |
| number of dark dorsal cross-bands | 5 | 5–7 | 5 | 5 | 4 | 5–7 | 6–7 | 6–7 | 5 | 5 | 5 | 5 |
| prominent pale supra-ocular bar | present | present | present | present | weak or absent | present | weak | weak | present | weak or absent | weak or absent | absent |
| width of light dorsolateral line (if present) versus vertebral line | broader | narrower | broader | similar | absent | broader | broader | broader | broader | narrower | absent | similar |
| belly with extensive dark patterning | eastern, variable | yes | no | no | no | variable | variable | variable | no | no | no | no |
| males and females strongly dimorphic | no | no | no | yes | no | no | no | no | no | northern | no | no |

**Diagnosis.** A species of *Tympanocryptis* with rostral scale continuous, or almost so, with the canthus rostralis, nasal scale below the canthus and not bordered below by enlarged scales, well-developed lateral neck fold, keeled dorsal head scales, usually a well-developed five-lined colour pattern in which the dorsolateral lines are equal to or broader than the vertebral line, no lateral skin fold, venter variable—immaculate to weakly to moderate brown flecks on throat.

**Description.** Lateral neck fold well developed, from angle of jaw to gular fold; spines along extent of fold. Head shape, moderately wide skull with moderately short snout. Head and snout with strongly keeled dorsal scales; keels irregular, those on the lateral scales aligned more obliquely than those on the more medial scales. Snout shape smoothly tapering in profile, the canthal scales continuous with the rostral scale. Nasal scale dorsal margin does not cross to the dorsal side of the canthus rostralis. No row of enlarged scales along the ventral margin of the nasal scale between the nasal and small snout scales. Dorsal body scales moderately to strongly keeled and imbricate. Numerous scattered enlarged spinous dorsal scales, at least twice the width of adjacent body scales, each with a strong median keel ending in a prominent spine directed posterodorsally, trailing edge of scale strongly raised. Enlarged spinous scales appear larger and more densely packed in the dark dorsal cross-bands than the paler interspaces, not arranged in longitudinal series. Ventral body scales smooth to weakly keeled, throat scales smooth to very weakly keeled. Thigh scalation heterogeneous, with scattered enlarged spinous scales similar to those on body. Lateral fold between axilla and groin absent. SVL to 59 mm in females and 54 mm in males.

Dorsal colour pattern variable in the degree of development and colour hue, from reddish brown to grey-brown with five dark brown transverse bands and with five-lined pattern variable, often well defined and continuous but often interrupted and expressed only where it crosses the dark cross-bands. Dorsolateral lines as wide as or wider than the vertebral line, well defined, straight edged, not expanding around the vertebral blotches. Vertebral and dorsolateral stripes continue weakly onto the tail. Pale supra-ocular bar usually strongly contrasting. Venter white, sometimes with weak to extensive dark peppering on the throat and, less commonly, the belly.

**Distribution.** Southern and eastern South Australia, from the Eyre Peninsula around Venus Bay in the west, north and east to the Lake Frome area, extending east to northwestern Victoria and far west of New South Wales [39].

**Variation.** Within populations across the range of *T. petersi*, there is local variation in the boldness of patterning, whether the dorsal lines are continuous or broken and how prominent the supra-orbital bar is on the head. Stronger patterning tends to be more typical of eastern and southern populations (electronic supplementary material, figure S1 in appendix S2), while the lined pattern is more frequently interrupted and contrast in pattern reduced in the northern and western parts of the range. Breeding male *T. petersi* have yellow colouring on the throat and the dorsolateral light lines also become suffused with yellow. Females are patterned as more muted versions of the patterns of the males with which they co-occur, and most have the dorsolateral light lines broken and the lateral lines only weakly evident. Ventral patterning shows some geographical variation, most populations being immaculate white on the belly and with some light grey speckling on the throat, but eastern animals can be quite strongly patterned all over the ventral surface with dark brown speckling (e.g. paratype NMV D74168 from Lake Tyrell).

**Comparison with other species.** *Tympanocryptis petersi* overlaps or potentially contacts a number of other *Tympanocryptis* species.

At its eastern limits, *T. petersi* is within 200 km of *T. pinguicolla.* The two differ in the absence of a lateral skin fold in *T. petersi* (present in *T. pinguicolla*), variable presence of weakly keeled ventrals (always smooth in *T. pinguicolla*) and consistent pattern of five dark transverse body bands (frequently six or seven in *T. pinguicolla*). *Tympanocryptis pinguicolla* occurs in higher rainfall grassland habitats, in contrast with the preference of eastern *T. petersi* for semi-arid salt lake and mallee habitats. *Tympanocryptis houstoni* approaches (but does not contact) *T. petersi* to the west, differing in its pale vertebral light line clearly broader than the dorsolateral light lines, consistent presence of moderate to heavy black ventral speckling and sporadic presence of six or seven dark dorsal cross-bands.

*Tympanocryptis petersi* and *T. argillosa* sp. nov. (Species B) approach each other in the area between Lake Frome and on the eastern border of South Australia. Sampling from this area is very sparse, but there again seem to be some mismatches of morphological and molecular markers indicating a level of contact and at least occasional hybridization. Two specimens that are phenotypically *T. argillosa* sp. nov. (SAMA R46040, R48353) have *T. petersi* mtDNA, indicating past hybridization events; both specimens are from populations hundreds of kilometres north of the nearest morphotypical *T. petersi*. Unlike the case with *T. tolleyi* sp. nov. (Species D, see below), checking the identity of specimens with RAG1 produced inconsistent results, with one of the mitochondrially anomalous specimens (R48353) scoring as *T. petersi*, in agreement with the mitochondrial data but in conflict with the morphology. Similarly, another specimen

(SAMA R54247) mitochondrially and morphologically typed as *T. argillosa* sp. nov. (Species B) was placed with *T. petersi* for RAG1. This specimen was included in the CT-scanned dataset and scores as an 'extreme' Species B with a particularly curved, deep snout. Overall, it is evident from the mitochondrial and nuclear DNA data that *T. petersi* and *T. argillosa* sp. nov. (Species B) are very recently diverged and there is evidence that some reticulate evolution occurred recently. Our recognition of the two as distinct species rests on a majority tendency of the DNA-typed specimens to accord with the morphology, notably snout shape and scalation, but this decision may need to be modified in future. At present, poor sampling in the putative overlap or contact zone roughly between Lake Frome and Lake Blanche prevents a more detailed inquiry into the nature and timing of genetic introgression between these two morphologically and biogeographically consistent entities.

*Tympanocryptis petersi* and specimens assigned to *T. tolleyi* sp. nov. (Species D) contact and narrowly overlap in the Gawler Ranges area of South Australia. All of the specimens from this area have *T. petersi* mtDNA, but most of the specimens in this region, running from Hiltaba in the west to Paney in the east, are unambiguously assignable to *T. tolleyi* sp. nov. (Species D) on morphology. Two of the morphologically problematic specimens were included in our limited RAG1 dataset, and both fell out with *T. tolleyi* sp. nov. (Species D). However, some males from a region running from Paney northwest to the area between Lake Acraman and the southern end of Lake Gairdner show a strongly developed vertebral light line and some females show better development of light dorsolateral lines over the dark cross-bands, both more like *T. petersi* than *T. tolleyi* sp. nov. (Species D). We believe this area represents a contact zone between the two, with some ongoing local gene flow in the eastern part of the Gawler Ranges.

**Habitat.** Inhabits a wide range of open warm temperate, semi-arid and arid habitats. In South Australia, it occurs in open grassland and chenopod shrublands, around salt lakes and claypans and in coastal dry scrubland, all habitats typically lacking a dense overstorey of trees (electronic supplementary material, figure S1 in appendix S2). In Victoria and southwestern New South Wales, it appears to be mainly restricted to salt lakes, such as Lake Tyrell. Much of its range has been heavily impacted by land-clearing and habitat fragmentation for agriculture, intensification of horticultural practice and changed water regimes along the Murray-Darling irrigation system.

**Remarks.** *Tympanocryptis petersi* is a new name applied to a species long known as *T. lineata*, a technicality enforced by the mistaken choice of a lectotype specimen that proved not to have come from the assumed type locality [19]. Conservation status of this species under its former name of *T. lineata* has been an issue across its distribution for some time, and is treated in more detail in Discussion.

**Etymology.** Named for Wilhelm Carl Hartwig Peters, pioneering German herpetologist and Director of the Zoological Museum of the University of Berlin from 1857 until his death in 1883. Peters coined the name *Tympanocryptis* and described its type species, *T. lineata*, based mainly on specimens from the Adelaide area. We take this opportunity to link his name to his discovery.

### *Tympanocryptis houstoni* Storr (1982)
Nullarbor Earless Dragon
(figures 9*b* and 11*b*)
Storr, G. M. (1982). *Records of the Western Australian Museum* 10(1): 61–66 (p. 62).
**Holotype**. WAM R53427, 10 km SSE Cocklebiddy, Western Australia (32°07′ S, 126°06′ E).

**Diagnosis.** A species of *Tympanocryptis* with rostral scale continuous with the canthus rostralis, nasal scale below the canthus and not bordered below by enlarged scales, well-developed lateral neck fold, keeled dorsal head scales, well-developed five-lined colour pattern in which the dorsolateral lines are narrower than the vertebral line and with sometimes more than five dark dorsal cross-bands, no lateral skin fold, venter heavily patterned with black flecks.

**Description.** Lateral neck fold well developed, from angle of jaw to gular fold; spines along the extent of fold. Head shape, moderately wide skull with moderately short snout. Head and snout with strongly keeled dorsal scales; keels irregular, those on the lateral scales aligned more obliquely than those on the more medial scales. Snout shape smoothly tapering in profile, the canthal scales continuous with the rostral scale. Nasal scale dorsal margin does not cross on to the dorsal side of the canthus rostralis. No row of enlarged scales along the ventral margin of the nasal scale between the nasal and small snout scales. Dorsal body scales strongly keeled and imbricate. Numerous scattered enlarged spinous dorsal scales, at least twice the width of adjacent body scales, each with a strong median keel ending in a prominent spine directed posterodorsally; trailing edge of scale strongly raised. Enlarged spinous scales appear larger and more densely packed in the dark dorsal cross-bands than the paler interspaces, not arranged in longitudinal series. Ventral body scales and throat scales smooth. Lateral fold between axilla and groin absent. SVL to 61 mm in females and 57 mm in males.

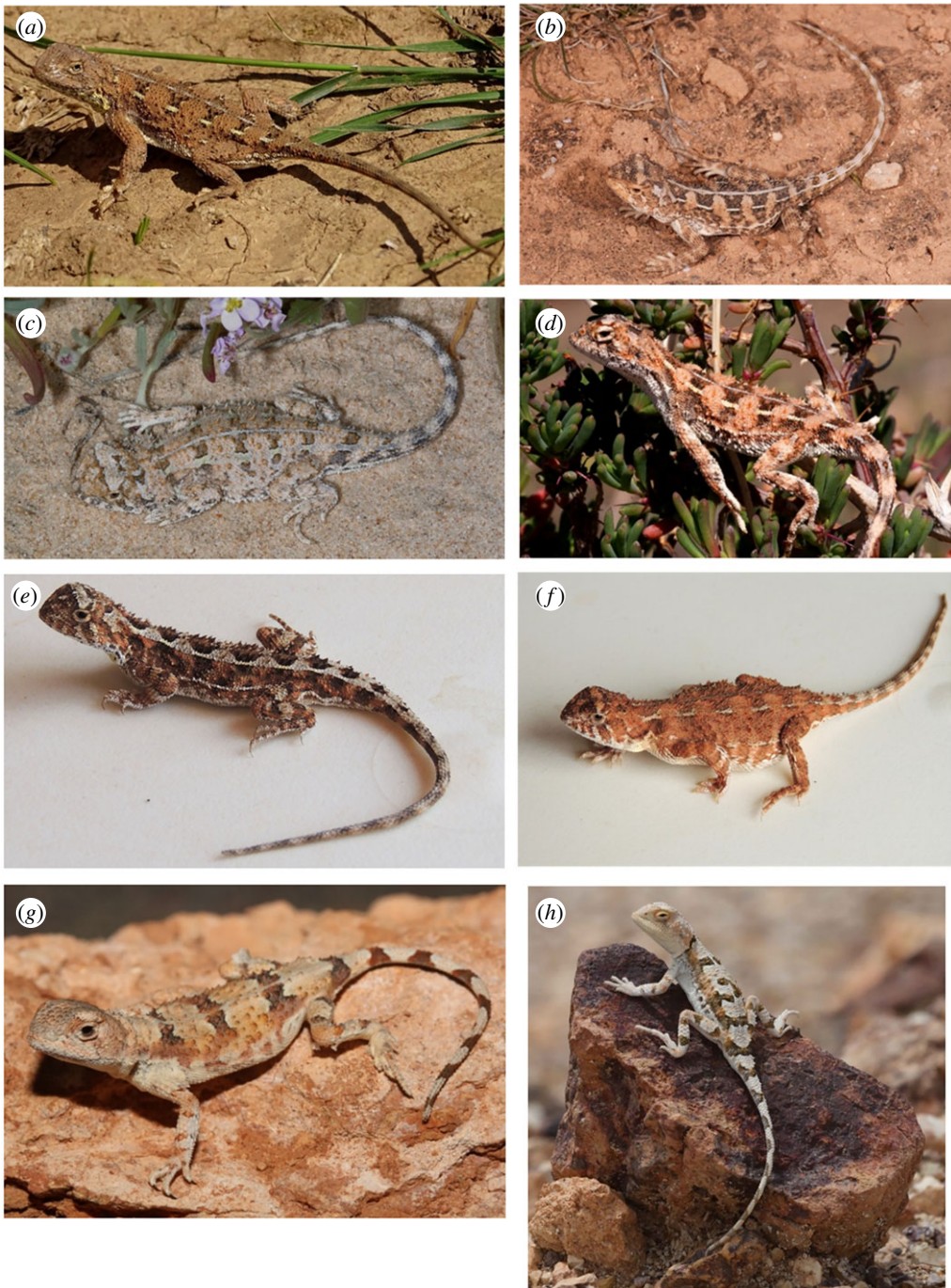

**Figure 9.** *Tympanocryptis* 'Southern' group species in life: (*a*) *T. petersi* sp. nov.—Male, Wandillah, South Australia (M. Hutchinson); (*b*) *T. houstoni*—Male, 100 km west of Eucla, South Australia (S. Macdonald); (*c,d*) *T. argillosa* sp. nov.—two animals, both from Moomba area, South Australia (S. Wilson); (*e,f*) *T. tolleyi* sp. nov.—two animals in life, male (left) and female (right) and (*g,h*) *T. fictilis* sp. nov.—two males in life, the Breakaways, South Australia (left, M Hutchinson; right, P. Tremul).

Dorsal colour pattern well developed and complex, sandy to medium brown dorsally with five dark brown transverse bands and with five-lined pattern either continuous or interrupted but always well defined. Vertebral pale line markedly wider than dorsolateral lines, the latter well defined, straight edged, not expanding around the vertebral blotches. Pale supra-ocular bar strongly contrasting and usually bordered posteriorly by a pale blotch above the jaw muscles. Vertebral and dorsolateral stripes continue onto the tail. Entire ventral surface heavily patterned by black speckles and blotches over a white background.

**Distribution.** The Nullarbor Plain and areas immediately adjacent to the west and east, from the Fraser Range in Western Australia [21] to the vicinity of Nundroo in South Australia.

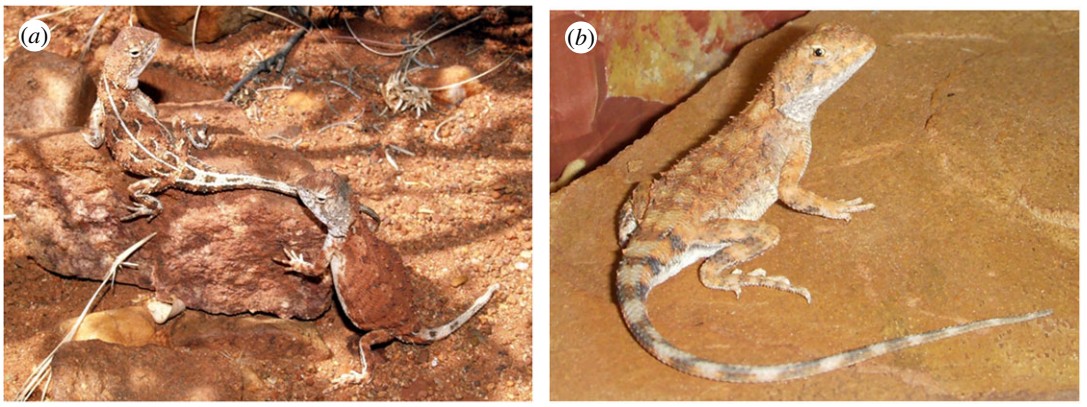

**Figure 10.** *Tympanocryptis* 'central pebble' group species in life: (*a*) *T. centralis*—male (left) and female, KataTjuta, Northern Territory (M Hutchinson) and (*b*) *T. rustica* sp. nov.—holotype (NMV D74671), male, Warrego Road, north of Tennant Creek, Northern Territory (J. Melville).

**Figure 11.** Holotypes: (*a*) *T. petersi* sp. nov.; (*b*) *T. houstoni*; (*c*) *T. argillosa* sp. nov.; (*d*) *T. tolleyi* sp. nov.; (*e*) *T. fictilis* sp. nov.; (*f*) *T. centralis* and (*g*) *T. rustica* sp. nov.

**Variation**. Variation occurs in the degree of continuity of the light dorsolateral stripes, with some specimens having the stripes only expressed where they transect the dark cross-bars (electronic supplementary material, figure S2 in appendix S2). There is not strong sexual dichromatism or pronounced sex-based differences of dorsal scalation in *T. houstoni*; as in *T. petersi*, females may be somewhat less contrasting in colour pattern and more often show interruption of the light dorsolateral and midlateral light stripes.

**Comparison with other species.** It is most similar to *T. petersi*, but distinguishable from both it and its other geographical near neighbour (see Remarks), *T. tolleyi* sp. nov. (Species D) by its broad vertebral stripe and moderate to intense black ventral markings.

**Habitat.** Inhabits mainly very open, generally treeless, samphire (*Tecticornia*) or chenopod (*Atriplex*, *Maireana*)-dominated plains ([40]; electronic supplementary material, figure S2 in appendix S2).

**Remarks**. Although Smith *et al.* [22], relying on distribution maps in Houston [41], assessed *T. houstoni* and *T. 'lineata'* (i.e. *T. petersi* plus Species D in the present study) as sympatric, this is not currently documented by confirmed specimens. At present, there are no records of any *Tympanocryptis* specimens in the 250 km hiatus between *T. houstoni* and *T. petersi* along the coastal hinterland from Nundroo to Venus Bay Conservation Park, nor further inland between Nundroo and Kycherring Rocks, the closest confirmed occurrence of *T. tolleyi* sp. nov. (Species D). A possible range overlap with Species D is suggested by an old specimen of *T. houstoni* said to be from Tarcoola (SAMA R1010), but the specimen has incomplete data and remains uncorroborated by any subsequent records.

*Tympanocryptis argillosa* sp. nov.
ZooBank urn:lsid:zoobank.org:act:636749F6-2F94-4C6D-949C-1856A0A47B9D
Claypan Earless Dragon
[Referred to as Species B in the Results section]
(figures 9*c*,*d* and 11*c*)

**Holotype.** SAMA R49360 E Side of Lake Hope Channel,12 km SSE Red Lake Yard, Strzelecki Desert, South Australia, 28°19′11″ S, 139°13′14″ E, adult female. Collected by M.H. on 9 April 1997.

**Paratypes.** NMV D3119, Lake Eyre, South Australia, 28°22′ S, 137°22′ E, adult female; SAMA R3619, Lake Palankarinna, South Australia, 28°46″ S, 138°25″ E, adult female; R46073, Artemia Point, Hunt Peninsula, Lake Eyre North, South Australia, 28°41″ S, 137°22″ E, adult male; R51039, 6.3 km NNW of Wimma Hill, South Australia, 27°17′32″ S, 140°17′26″ E, adult male; R51360–61, 4.4 km NNW of Beal Hill, Simpson Desert Regional Reserve, South Australia, 26°35′43″ S, 137°49′51″ E, adult male; R54088, 4.1 km SW of Kalamurina homestead, South Australia, 27°55′36″ S, 137°57′22″ E, adult male: R54247, 23.5 km W of Kannakaninna Waterhole, Kalamurina Station, South Australia, 27°53′40″ S, 137°44′23″ E, adult female; R54248, 23.5 km W of Kannakaninna Waterhole, Kalamurina Station, South Australia, 27°53′40″ S, 137°44′23″ E, adult male.

**Diagnosis.** A species of *Tympanocryptis* with rostral scale separated from the canthus rostralis, nasal scale extending dorsally across the canthus and bordered below by enlarged scales, well-developed lateral neck fold, no lateral skin fold, weakly keeled dorsal head scales, five-lined pattern present but light lines usually discontinuous, ventral surface white.

**Description.** Lateral neck fold weakly continuous; spines not continuous along fold, with cluster at jaw angle and then forming a line along the posterior end of fold. Head shape, wide skull with short snout. Head and snout with moderately keeled dorsal scales; keels irregular, those on the lateral scales aligned more obliquely than those on the more medial scales. Snout shape convex in profile, with one to two rows of supralabial scales separating the rostral area from the canthus rostralis. Nasal scale dorsal margin extends on to the dorsal side of the canthus. Two to three enlarged scales along the ventral margin of the nasal scale, between the nasal and small snout scales. Dorsal body scales weakly keeled or flat, only slightly overlapping. Scattered enlarged dorsal scales, at least twice the width of adjacent body scales, each with a strong median keel ending in a small posterior spine directed posterodorsally, very narrow slightly raised trailing edge. Enlarged spinous scales appear larger and more densely packed in the dark dorsal cross-bands than the paler interspaces, not arranged in longitudinal series. Ventral body scales and throat scales smooth. Thigh scalation heterogeneous, with scattered enlarged spinous scales similar to those on body. Lateral fold between axilla and groin absent. SVL to 47 mm in females and 52 mm in males.

Dorsal colour pattern often weakly contrasting, pale ochre to grey-brown with five somewhat darker transverse bands, and with five-lined pattern interrupted, the vertebral and dorsolateral stripes expressed only where they cross the dark dorsal blotches; striped pattern not continuing on to tail. Pale supra-ocular bar moderate to strongly contrasting. Venter white.

**Distribution.** Throughout the Strzelecki Desert, South Australia, and southern Simpson Desert in northeastern South Australia south to Lake Frome, far southwestern Queensland and far western New South Wales as far south as the Fowlers Gap region.

**Variation.** The intensity of patterning on dorsal surface varies somewhat between individuals, with males tending to be more strongly patterned, while females tend to have less distinct patterning, with fainter, less continuous dorsolateral lines, but the degree of difference between the sexes is slight compared with some other species treated here (electronic supplementary material, figure S3 in appendix S2). Unknown whether breeding males develop bright breeding colouring.

**Comparison with other species.** The morphological and molecular data show a mismatch in the area around Olympic Dam, central South Australia where a specimen genotyped as *T. argillosa* is phenotypically *T. tolleyi* sp. nov. (Species D) (SAMA R37888; electronic supplementary material, figure S5 in appendix S2). This suggests earlier contact or long distance dispersal of *T. argillosa* to the Olympic Dam region of South Australia. In this region, all animals examined are phenotypically *T. tolleyi* sp. nov. (Species D), suggesting capture of the *T. argillosa* mtDNA haplotype. There is no evidence that *T. tolleyi* sp. nov. (Species D) and *T. argillosa* are currently sympatric or closely allopatric, the nearest records being separated by over 200 km.

In the southeastern parts of its range, *T. argillosa* overlaps with its sister species *T. petersi*, with evidence of some hybridization. In the contact zone, *T. argillosa* can be distinguished from *T. petersi* by its pale creamy brown to beige body colour with little patterning (versus richer red-brown colour and bolder patterning), broken poorly defined dorsolateral lines only visible in the darker cross-bands (versus more continuous and well-defined dorsolateral lines), presence of flat unkeeled dorsal body scales (versus all body scales being keeled and imbricate) and two to three enlarged scales along the ventral margin of the nasal scale (versus no enlarged scales with small lateral snout scales contacting the nasal scale).

*Tympanocryptis argillosa* overlaps with *T. tetraporophora*, from which it can be distinguished by having a well-developed neck skin fold, and no femoral pores (versus two present), and *T. intima*, from which it can be distinguished by the presence of the neck skin fold, keeled (rather than smooth) dorsal head and snout scales and irregularly scattered enlarged dorsal tubercules (versus a few enlarged tubercules forming longitudinal series).

**Habitat.** Poorly known. Specimens mostly associated with the clayey swales between the dunes of the sandy deserts where it occurs (electronic supplementary material, figure S3 in appendix S2).

**Etymology.** Adjectival form of the Latin word for clay, *argilla*, alluding to the preference of this species for the heavier soiled clay swales that are commonly encountered between the dunes of the Strzelecki and southern Simpson Deserts.

*Tympanocryptis tolleyi* sp. nov.
ZooBank urn:lsid:zoobank.org:act:EA3D1AFA-9E8B-46EE-A02A-B8C4681EC1AF
Gawler Earless Dragon
[Referred to as Species D in the Results section]
(figures 9*e,f* and 11*d*)

**Holotype.** SAMA R69147, 2.8 km ESE Malbooma Outstation, Mulgathing Station, South Australia; 30°41′47″ S, 134°12′21″ E; adult male. Collected by M.H. on 13 October 2015.

**Paratypes.** SAMA R26711, 30 km SW Mabel Ck H/S, South Australia, 29°10′30″ S, 134°08′ E, adult female; R39978-79, 2 km SSW of Grace Bore, Bulgunnia Station, South Australia, adult and juvenile females; R44704, Mt Vivian Yards, The Twins Station, South Australia, 30°36′ S, 135°42′ E, adult female; R45143, Nth Halls Paddock, Wilgena Station, South Australia, 30°35′03″ S, 134°19′37″ E, adult male; R47558, 1.9 km NE of Crows Nest Bore, Mt Vivian Station, South Australia, 30°49′22″ S, 135°49′18″ E, adult male; R58048, 25 km NE Half Moon Lake, South Australia, 29°52′34″ S, 133°37′05″ E, adult male; R62565, Challenger Mine lease, 23.6 km SE Blowout Trig, South Australia, 29°52′32″ S, 133°34′50″ E, adult female; R65637, 8.4 km NE Bon Bon Homestead, South Australia, 30°21′49″ S, 135°31′45″ E, adult male; R65808, 5.12 km NE Mullaquana Homestead, 33°10′38″ S, 137°24′08″ E, adult male; R69085, 0.2 km SSW Malbooma Outstation, Mulgathing Station, South Australia, 30°41′17″ S, 134°10′40″ E, adult female; R69141, 1.1 km SE Malbooma Outstation, Mulgathing Station, South Australia, 30°41′38″ S, 134°11′10″ E, adult male.

**Diagnosis.** A species of *Tympanocryptis* with rostral scale separated from the canthus rostralis, nasal scale extending dorsally across the canthus and bordered below by enlarged scales, well-developed lateral neck fold, keeled dorsal head scales, strong sexual dimorphism in dorsal pattern (cross-banded females, males with longitudinal light dorsal lines that coalesce medially), ventral surface white.

**Description**. Lateral neck fold weakly continuous; spines not continuous along fold, with cluster at jaw angle and then forming a line along the posterior end of fold. Head shape, moderately wide skull with moderately short snout. Head and snout with moderately keeled dorsal scales; keels irregular, those on

the lateral scales aligned more obliquely than those on the more medial scales. Snout shape convex in profile, with one to two rows of supralabial scales separating the rostral area from the canthus rostralis. Nasal scale dorsal margin extends on to the dorsal side of the canthus. Two to three enlarged scales along the ventral margin of the nasal scale, between the nasal and small snout scales. Dorsal body scales weakly keeled or flat, only slightly overlapping. Numerous scattered enlarged dorsal scales, at least twice the width of adjacent body scales, each with a strong median keel ending in a prominent posterior spine directed posterodorsally; trailing edge of scale with narrow raised posterior edge. Enlarged spinous scales appear larger and more densely packed in the dark dorsal cross-bands than the paler interspaces, not arranged in longitudinal series. Ventral body scales smooth to weakly keeled, throat scales weakly keeled. Thigh scalation heterogeneous, with scattered enlarged spinous scales similar to those on body. Lateral fold between axilla and groin absent. SVL to 55 mm in females and 50 mm in males.

Males and females markedly different in pattern. Males dorsally tan, sandy orange or pinkish brown with prominent whitish dorsolateral lines that are usually continuous and widen medially between the dark dorsal cross-bands, outlining a series of five isolated dark patches down the vertebral zone. Females drabber grey-brown to sandy brown, with almost no trace of longitudinal lines, leaving a simple dorsal body pattern of five irregular dark transverse dorsal cross-bands, somewhat narrower than the intervening pale body colour. Pale vertebral stripe in both sexes tends to be narrow and weakly expressed, usually confined to dark mid-dorsal patches. The dorsolateral striped pattern continues strongly onto tail in males. Pale supra-ocular bar moderate to strongly contrasting. Venter white.

**Distribution.** South-central South Australia, from about 133° E longitude (vicinities of Mt Igy and Wynbring), southeast to Whyalla and east to the Roxby Downs area. This area corresponds closely to the Gawler Bioregion [42].

**Variation.** The extent of patterning on dorsal surface varies geographically, with northwestern animals having males with more discontinuous dorsolateral lines, although the dimorphic pattern of dark vertebral blotches surrounded by whitish is still present (electronic supplementary material, figure S4 in appendix S2). Pale supra-orbital bar on the top of head variable from prominent to faintly visible. Variation in the amount of pigmentation on the ventral surface ranges from no pigmentation through to dark mottling on chin and throat. Males do not show bright yellow to red breeding colouring. The distinctive male colour pattern of *T. tolleyi* was previously documented on the Eyre Peninsula as the 'Upper Eyre Peninsula race' of *T. 'lineata'* [43, pp. 46–47, 78].

As noted above, there is a region of possible overlap between *T. tolleyi* and *T. petersi* on the upper Eyre Peninsula and eastern Gawler Ranges (electronic supplementary material, figure S5 in appendix S2). Further research is needed in the eastern Gawler Ranges to determine the level of genetic exchange between species and differences in habitat associations and ecological requirements of each.

As mentioned above, there is evidence of mtDNA introgression between *T. tolleyi* and *T. argillosa* in the Olympic Dam region. This evidence is based on a combination of mtDNA, nuclear (RAG1) DNA and morphology (electronic supplementary material, figure S5 in appendix S2). Further research is needed to determine whether this is historical gene capture of the mtDNA haplotype.

*Tympanocryptis tolleyi* also occurs in close allopatry to its mtDNA sister species *T. fictilis* sp. nov. (Species C) from which it can be readily distinguished by having five (versus four) dark cross-bands or dorsal blotches on the body. It also overlaps with the distribution of *T. tetraporaphora*, from which it can be distinguished by having a strong neck skin fold, pale supra-orbital bar and no femoral pores, and *T. intima*, from which it can be distinguished by the presence in *T. tolleyi* of a strong neck skin fold, lightly to strongly keeled dorsal snout scales, strongly contrasting cross-bands or dorsal blotches on body, and randomly scattered enlarged dorsal tubercules (versus forming longitudinal series in *T. intima*).

**Habitat.** Arid chenopod shrublands and grasslands and the skirts of stony hills and rock outcrops, generally with sparse overstorey and with gravelly or heavier soil types (electronic supplementary material, figure S4 in appendix S2). In the vicinity of the type locality at Malbooma outstation and the Kychering Rocks, lizards inhabit both the sparsely vegetated granitic and sedimentary rock outcrops and the surrounding gravel and clay plains.

**Etymology.** Named for Timothy David Tolley of the South Australian Museum Waterhouse Club, in recognition of his long-term support for field research by the South Australian Museum, including coordination of the Waterhouse Club expedition that yielded the series of specimens that includes the holotype of this species.

**Tympanocryptis fictilis** sp. nov.
ZooBank urn:lsid:zoobank.org:act:EEE8BF54-4D5B-4BD5-99D5-87A0C8132171
Harlequin Earless Dragon

[Referred to as Species C in the Results section]

(Figures 9*g*,*h* and 11*e*)

**Holotype.** SAM R46179 18 km NNE of Arckaringa Homestead, South Australia, 27°46′51″ S, 134°47′11″ E, adult male. Collected by Ralph Foster and Sharon Downes, 30 September 1995.

**Paratypes.** SAMA R44707, Douglas Dam Track, 4 km S Eucalyptus Waterhole, 27°36′46″ S, 134°35′59″ E, adult female; R46226, Nr Douglas Dam, Arckaringa, South Australia, 27°39′7″ S, 134° 32′55″ E, adult female; R46228, 17 km NNE of Arckaringa Homestead, approximately 50 km SW of Oodnadatta, South Australia, 27°47′38″ S, 134°47′13″ E, adult male; R44647, 4 km S of Eucalyptus Waterhole, Douglas Dam Track, approximately 50 km W of Oodnadatta, South Australia; 27°36′46″ S, 134°35′59″ E, adult male; R44712, 1 km S of Eucalyptus Waterhole, Todmorden Station, approximately 50 km west of Oodnadatta, South Australia, 27°35′09″ S, 134°36′36″E, adult female; R48493-94, 4.1 km Ese Parke Camp Waterhole, Todmorden Station, South Australia, 27°20′49″ S, 134°29′23″ E, adult females; R58134-35, W of Stuart HWY, 18.3 km WSW of England Hill, 80 km N of Coober Pedy, South Australia, 28°10′27″ S, 134°01′49″ E, adult males; R58187, 16.7 km WNW of Pile Hill, South Australia, 28°45′01″ S, 134°32′21″ E, adult male.

**Diagnosis.** A species of *Tympanocryptis* with smooth head and snout scales, rostral scale separated from the canthus rostralis, no lateral neck fold, smooth dorsal head scales, simple dorsal colour pattern of four dark dorsal cross-bands and no longitudinal five-lined pattern apart from traces of a vertebral line, ventral surface white.

**Description.** Lateral neck fold not developed. Head shape, very wide skull with very short snout. Head and snout with smooth or very weakly keeled dorsal scales. Snout shape convex in profile, with one to two rows of supralabial scales separating the rostral area from the canthus rostralis. Nasal scale dorsal margin extends on to the dorsal side of the canthus. Two to three enlarged scales along the ventral margin of the nasal scale, between the nasal and small snout scales. Dorsal body scales flat, unkeeled, not imbricate. Scattered enlarged dorsal scales, at least twice the width of adjacent body scales, each with a slightly raised central keel ending in a small spine and with a raised trailing edge; most enlarged scales also have two dark spots of pigmentation on trailing edge, either side of central keel. Enlarged spinous scales uniformly scattered across both dark dorsal cross-bands and paler interspaces, not arranged in longitudinal series. Ventral body scales and throat scales smooth. Lateral fold between axilla and groin absent. Thigh scalation heterogeneous, with scattered enlarged spinous scales similar to those on body. SVL to 51 mm in females and 49 mm in males.

Dorsal colour pattern pale cream to pale brown with four strongly contrasting but narrow darker transverse bands, and with lined pattern absent except for a discontinuous vertebral line in some individuals that is only visible on dark cross-bands; no longitudinal striped pattern on tail. Pale supra-orbital bar on the top of head very weak or absent. Lateral fold between axilla and groin absent.

**Distribution.** A narrow strip of central northern South Australia, from about Coober Pedy north to the Alberga Creek, apparently bounded by sand dune deserts in the north (Pedirka Desert) and west (Great Victoria Desert). Current easternmost records are from Arckaringa Station (to about 135 E), but eastern range limits uncertain due to lack of sampling and may extend as far as the Simpson Desert.

**Variation.** The width and extent of the four dark dorsal cross-bars varies between individuals (electronic supplementary material, figure S6 in appendix S2), from narrow continuous bars (SAMA R46178) to broad bars that form large blobs between broken dorsolateral stripes (SAMA R58134). However, even in animals with broad dark cross-bands, they are narrower than intervening pale background colour. Most individuals lack pale supra-orbital bar on top; however, two animals examined had wide poorly defined pale supra-orbital bar (SAMA R67258, R58135). Variation in the amount of pigmentation on the ventral surface ranges from no pigmentation through to dark mottling on head, throat and lateral portions of body. Most individuals have some mottling on the head and throat, including males and females. Breeding male *T. fictilis* have a yellow wash on the throat and along the lateral edges of the venter. Compared with females, male body patterns tend to be more intensely contrasting, the pale zones being almost white in some animals, while the dark cross-bands are very dark umber brown.

**Comparison with other species.** Very distinctive, *T. fictilis* is readily distinguished by its smooth head and body scales, lack of a lateral neck fold (thereby having a very distinct neck) and having four (rather than five) strongly contrasting dorsal cross-bands on the body, with the dark cross-bands narrower than intervening pale background colour (versus five dark cross-bands or pale-outlined dark dorsal blotches as wide as the pale interspaces). Specimens of this species were first noted by Houston [43] as *T. cephalus*, because it lacks several characters that are typical of *T. lineata sensu lato*, such as the neck fold, strongly keeled head scales and keeled dorsal scales. In all of these respects, it resembles the *T. cephalus* complex and *T. intima* much more than its close relatives *T. tolleyi* and *T. petersi*. It occurs in near allopatry to its

mitochondrial sister species *T. tolleyi*, current records separating them by about 50 km. The distribution of *T. fictilis* overlaps with the that of *T. intima*, from which it can be distinguished by having strongly contrasting dorsal cross-bands and enlarged dorsal tubercules scattered over the back (versus forming longitudinal series), and *T. tetraporaphora*, from which it can be distinguished by having smooth head and body scales and two pre-anal pores (versus keeled scales and two additional femoral pores).

**Habitat.** Very open, often undulating, stony plains, including areas of low rock outcrops with sparse chenopod ground cover and scattered *Acacia* overstorey (electronic supplementary material, figure S6 in appendix S2). Through much of this area, a common component of the ground surface is silcrete gravel, its smooth surfaces and white to pale beige colouring contrasting with the underlying light to dark reddish clay. The colour and pattern of *T. fictilis* along with its smooth-surfaced scales provide a remarkably effective camouflage against this clay-silcrete land surface.

**Etymology.** From *fictilis*, Latin adjective meaning fashioned from clay, as in porcelain or pottery, referring to the smooth scale surface and strongly contrasting colours of this lizard that match the equally contrasting silcrete and clay substrate on which they live.

### *Tympanocryptis centralis* Sternfeld, 1924
Central Pebble Dragon.
(figures 10*a* and 11*f*)

*Tympanocryptis lineata centralis* Sternfeld, R. 1924. *Abhandlungen der Senckenbergischen Naturforschenden Gesellschaft* 38: 221–251 (p. 234).

**Lectotype**. SMF 10349, Hermannsberg Mission, upper Finke River, Northern Territory. Collector M. Leonhardi (1908).

**Diagnosis.** A species of *Tympanocryptis* with rostral scale separated from the canthus rostralis, nasal scale extending dorsally across the canthus and bordered below by enlarged scales, no lateral neck fold, dorsal head scales keeled, enlarged dorsal spinous scales roughly aligned longitudinally, ventral and throat scales keeled, colour pattern geographically variable, from strongly five-lined and weakly dimorphic in the south to more weakly patterned and with stronger sexual dimorphism in the north, ventral surface white with weak throat speckling.

**Description.** Lateral neck fold absent. Head shape, narrow skull with long tapered snout. Head and snout with moderately keeled dorsal scales; keels irregular, those on the lateral scales aligned more obliquely than those on the more medial scales. Snout shape convex in profile, with a row of supralabial scales separating the rostral area from the canthus rostralis. Nasal scale dorsal margin extends on to the dorsal side of the canthus. Two to three enlarged scales along the ventral margin of the nasal scale, between the nasal and small snout scales. Dorsal body scales weakly keeled and imbricate. Numerous longitudinally oriented series of enlarged spinous scales that are twice the width of adjacent body scales, wider than long, with obvious raised posterior edges and prominent central keel. Ventral body scales weakly keeled, throat scales keeled. Thigh scalation heterogeneous, with scattered enlarged spinous scales similar to those on body. Lateral fold between axilla and groin absent. SVL to 55 mm in females and 54 mm in males

Dorsal light to dark sandy brown with five dark brown transverse bands. Dorsolateral light bands coalesce on the tail to produce a pale dorsal stripe, interrupted by dark dorsal blotches. The degree of expression of the five-lined pattern varies geographically (see Variation) from strongly developed to almost absent. Females often with a greyish 'cap' covering the dorsal surface of the head over the occipital region. Pale supra-ocular bar weak to almost absent. Venter white, with weak to obvious dark speckling, most evident on the throat.

**Distribution.** Rocky central ranges of southern Northern Territory from the northern edge of the West Macdonnell Ranges, south through Kings Canyon National Park, Kata Tjuta, and to the Indulkana, Musgrave and Tomkinson ranges in far northwestern South Australia and adjacent Western Australia.

**Variation.** In the rocky ranges of the northwest of South Australia and adjacent Western Australia and the Northern Territory, most individuals consistently have a five-lined pattern in which the pale lines are well defined and contrasting, although usually discontinuous over the paler interspaces between the dark dorsal blotches, and with little difference in pattern between males and females (electronic supplementary material, figure S7 in appendix S2). Animals around Kata Tjuta and Uluru in the southwest of the Northern Territory are strongly dimorphic; in males, the longitudinal lined patterning is always present, with continuous and prominent pale vertebral line and the dorsolateral lines variably continuous or interrupted. Females from this area are often almost unpatterned, with the dark dorsal bands only feebly evident and no or only faintly contrasting development of the vertebral and dorsolateral pale lines on the body, although the tail pattern is usually still evident.

**Comparison with other species.** No other *Tympanocryptis* species co-occur with *T. centralis*. Distinguished from its close relative Species E, which occurs further north, by strongly patterned males and usual presence of keels on the throat scales. The species that abuts the range of *T. centralis* most closely is *T. tetraporophora*, which can be superficially similar in having well-patterned males and often almost patternless females. *Tympanocryptis centralis* is distinguished by lacking a pair of femoral pores, by a pale transverse supra-orbital bar (usually discernible) and by the dorsal tubercles tending to be arranged in longitudinal series.

**Habitat.** Occurs on arid stone deserts, stony slopes skirting rocky hills or loams with scattered stones and spinifex grass (*Triodia* spp.) associated with desert ranges (electronic supplementary material, figure S7 in appendix S2).

**Remarks.** This species was part of the *T. lineata* species complex, but more recent molecular work demonstrated it is related to the pebble earless dragons, such as *T. cephalus* and *T. intima* [44]. The current work now redefines the distribution of *T. centralis*, restricting it to the southern parts of the Northern Territory, northwestern South Australia and adjacent far eastern Western Australia.

*Tympanocryptis rustica* sp. nov.
ZooBank urn:lsid:zoobank.org:act:2E0A1B50-510D-4516-BE9E-7D9A4C0B244C
Tennant Creek Pebble Dragon.
[Referred to as Species E in the Results section]
(figures 10*b* and 11*g*)

**Holotype.** NMV D74671, Warrego Road, N of Tennant Creek, 3 km W of Stuart Highway, Northern Territory, 19°35′57″ S, 134°09′37″ E, adult male. Collected by J. Melville and A. O'Grady, September 2009.

**Paratypes.** NMV D74688, Warrego Road, N of Tennant Creek, approximately 12 km W of Stuart Highway, Northern Territory; 19°33′31″ S, 134°06′12″ E, female; D74670, D74672, D74684, Warrego Road, N of Tennant Creek, 3 km W of Stuart Highway, Northern Territory, 19°35′57″ S, 134°09′37″ E, adult males; D74693-94, Binns Road, 1 km W of turnoff to Whistleduck Creek, Northern Territory, 20°28′18″ S, 134°46′50″ E, adult males; SAMA R38823, Tennant Creek Rubbish Dump, Tennant Creek, Northern Territory, 19°39′ S, 134°11′ E, female; R53990, Tennant Creek Rubbish Dump, Tennant Creek, Northern Territory, 19°39′35″ S, 134°10′35″ E, juvenile.

**Diagnosis.** A species of *Tympanocryptis* with rostral scale separated from the canthus rostralis, nasal scale extending dorsally across the canthus and bordered below by enlarged scales, no lateral neck fold, keeled dorsal head scales, enlarged dorsal spinous scales roughly aligned longitudinally, keeled ventrals but smooth throat scales and both sexes weakly patterned or patternless dorsally, and white ventrally.

**Description.** Lateral neck fold absent. Head shape, moderately wide skull with moderately long tapered snout. Head and snout with moderately keeled dorsal scales; keels irregular, those on the lateral scales aligned more obliquely than those on the more medial scales. Snout shape convex in profile, with a row of supralabial scales separating the rostral area from the canthus rostralis. Nasal scale dorsal margin extends on to the dorsal side of the canthus. Two to three enlarged scales along the ventral margin of the nasal scale, between the nasal and small snout scales. Dorsal body scales weakly to moderately keeled and imbricate. Sparsely scattered enlarged dorsal scales forming weak longitudinal series, at least twice the width of adjacent body scales, each with a weakly defined median keel ending in spine directed posterodorsally, trailing edge of scale not raised into a rim. Ventral body scales weakly keeled, throat scales keeled. Thigh scalation heterogeneous, with scattered enlarged spinous scales similar to those on body. Lateral fold between axilla and groin absent. SVL to 59 mm in females and 52 mm in males.

Dorsal reddish brown to greyish brown. Five somewhat darker brown transverse bands and narrow greyish dorsal lines may be discernible, but are only ever weakly contrasting and often virtually absent. Light dorsolateral lines more evident on the tail, producing a pattern interrupted by dark caudal blotches. Pale supra-ocular bar weak to almost absent. Venter white, with weak or no dark gular speckling.

**Variation.** A very plain-coloured species, and the few specimens examined show little variation (electronic supplementary material, figure S8 in appendix S2). Some males show the five-lined pattern, but the contrast between the pale lines and the background colour is low. The two adult females completely lack lines, while the males all show traces of them. Dorsal darker patches are only weakly evident, slightly darker than the background colour.

**Comparison with other species.** *Tympanocryptis rustica* is closely related to and most similar to *T. centralis*. It differs in its more extremely reduced patterning, and smoother throat scalation. It overlaps with *T. tetraporophora*, from which it can be distinguished by its lack of femoral pores and snout morphology, the rostral not contacting the canthus rostralis in *T. rustica*, but smoothly contacting it in *T. tetraporophora*

**Distribution.** Distributed in stone deserts of central Northern Territory, known from Tennant Creek in the north to as far south as the Iytwelepenty/Davenport Ranges National Park.

**Habitat.** Occurs on arid stone deserts or loams with scattered stones and spinifex grass (*Triodia* spp.) associated with desert ranges (electronic supplementary material, figure S8 in appendix S2). Has been observed perching on small stones in the open and fleeing into nearby spinifex grass when alarmed. Will also freeze in the open when alarmed, its appearance closely matching and blending with the pebbles-strewn ground surface in a similar manner to other members of the pebble earless dragon group.

**Etymology.** From *rustica*, Latin adjective meaning simple or plain, alluding to the uniform coloration of these weakly patterned dragons. The word is also an echo of the English word rusty that also describes the predominant dorsal colour in life.

### *Tympanocryptis macra* Storr (1982)
Savannah Earless Dragon

*Tympanocryptis lineata macra*. Storr, G.M. (1982). *Records of the Western Australian Museum* 10(1): 61–66 (p. 61).

**Holotype**. WAM R44553, 16 km S of main dam at Lake Argyle, Western Australia (16°15′ S, 128°40′ E). Collected by L.A. Smith and R.E. Johnstone, 20 January 1972.

**Status.** Our molecular work has revealed genetic structure within this species that warrants further investigation, but available samples have been insufficient to fully explore geographical variation within this taxon. At this time, we suggest continuing with Storr's concept of the taxon, but recognizing that this is a species, not a geographical variant of *T. lineata* (=*T. petersi*). Our data make it clear that *T. macra* is a distinct species, unrelated to the *lineata* complex and the most basally branching species in the genus *Tympanocryptis*.

**Diagnosis.** A species of *Tympanocryptis* with rostral scale continuous with the canthus rostralis, no lateral neck fold, keels of dorsal snout and head scales all aligned parallel to the midline, weak five-lined colour pattern, keeled ventrals and throat scales, and large size (to 64 mm SVL).

**Comparison with other species.** *Tympanocryptis macra* is a distinctive species, the largest species in the genus and the most gracile, with relatively longer extremities [21]. Its weakly contrasting colour pattern, as Storr [21] noted, makes it superficially similar to *T. rustica*, but its tapered snout and cephalic keels oriented in parallel series immediately distinguish *T. macra* from either *T. centralis* or *T. rustica*. Recent publications (e.g. [45]) have at times confused this species with *T. uniformis*, a species that is currently poorly understood, but markedly distinct from *T. macra* in its round-headed, 'dumpy' habitus and dorsal surface lacking enlarged spinous scales.

**Distribution.** Widely distributed across the northern edges of arid zone from the Kimberley region in Western Australia to mid-way across the Northern Territory.

**Habitat.** A wide range of tropical savannah grasslands and shrublands, mainly on cracking soils. Males observed perching on low rocks or clumps of soil using tripod stance, similar to that observed previously in *T. tetraporophora*. Will flee down cracks when alarmed. Has also been observed perched off the ground in grasses during the heat of the day.

## 4. Discussion

Our study integrated phylogeographic, phylogenetic, external morphological and internal osteological data, providing taxonomic support for seven already described *Tympanocryptis* taxa and strong evidence for five undescribed species, for which we provide a taxonomic treatment. Overall, we found that genetic lineages differed significantly in cranial shape, with significant pairwise differences between multiple lineages. Our results demonstrated that even a small ($n = 5$) sample from each genetic lineage could identify individuals for whom cranial shape did not match with the mtDNA lineage (e.g. SAMA R29021, R37888), but correlated with external morphology and the RAG1 nuclear marker. We were thus able to uncover evidence of mtDNA introgression at three contact zones. More generally, our quantification of morphological variation as part of whole-evidence taxonomy offered greater confidence in the existence of separate species across these cryptic lineages. Our study provides foundational evidence of hybridization and incomplete lineage sorting, where additional sampling and genetic data (e.g. genomics) are needed to explore this topic in more detail.

In this study, we addressed the taxonomic uncertainty of species that had, at one time or another, been included under the name *T. lineata*. Some morphological characters used in past taxonomic assessments to unite species under this taxon are not useful in species differentiation, and appear to be a case of

evolutionary convergence or retained ancestral traits. In particular, the previously used character combinations, including lack of femoral pores and dorsal patterns, do not appear to be useful in defining species within *Tympanocryptis*. We provide characters useful for distinguishing among the species of the '*T. lineata*' complex identified (table 5).

Knowledge of species diversity within *Tympanocryptis* continues to improve, with α-diversity in this genus increasing from 8 [45] to 20 species in the last 5 years alone [19,29,30]. These increases have occurred across all environments (e.g. arid, semi-arid, tropical and temperate) and habitats (e.g. stony deserts, cracking soils, chenopod shrublands, and temperate and tropical grasslands), allowing a broader perspective on macro-evolutionary patterns in these small cryptic lizards.

We observed that geometric morphometric analyses of cranial micro-CT scans and qualitative assessment of external head characteristics uncovered differences in length and width of the skull that are linked to their distribution in either the arid interior or coastal regions of Australia (figures 1*d* and 4*b*). Species from the interior of the continent, such as *T. centralis*, *T. rustica*, *T. argillosa*, *T. fictilis* and *T. tolleyi*, have short, wide snouts giving them a distinctive blunt head shape. These species are not sister taxa, occurring across two evolutionary lineages, and it is notable that members of completely separate iguanian radiations, the Palaearctic *Phrynocephalus* agamids [46] and the North American phrynosomatid 'sand lizards' (the *Uma-Phrynosoma* clade, [47]) that inhabit similar open, arid habitats, have a similar short, wide snout morphology.

## 4.1. Conservation implications

The recent taxonomic revision of the grassland earless dragons [19] highlighted the conservation plight of these species. These small cryptic grassland specialists appear to be particularly vulnerable to habitat loss and fragmentation, and all four GED species are of high conservation concern. In the current study, one particular species, *Tympanocryptis petersi*, also warrants assessment of its conservation status.

*Tympanocryptis petersi*, described in the current study, is the taxon that was referred to as *T. lineata* prior to nomenclatural review in Melville *et al.* [19] and as such is currently listed as occurring in all Australian mainland states [48]. Under our taxonomic revision, *T. petersi* (referred to as Species A in our Results section) is now restricted to southern grasslands and chenopod shrublands in northwestern Victoria, southeastern South Australia and southwestern New South Wales (figure 1*d*). Many of the locality points on our distribution map (figure 1*d*) are from historic museum samples with no recent specimens or sightings over the last 30–40 years. There have been significant declines in this species throughout its range, and although it is relatively common on the southern Eyre Peninsula in South Australia, there are only a few areas to the east where the species is known to survive. For example, in South Australia in the area of its original discovery (the Adelaide-Mt Lofty region), it is regarded as endangered [49]. In northwestern Victoria, this species occurs mainly in chenopod shrublands, usually associated with salinas or gypseous areas, where habitat losses and vegetation changes in response to clearing and herbivore grazing have severely impacted these environments (P. Robertson 2019, personal communication). *Tympanocryptis petersi* (formerly *T. lineata*) is now known from only five sites in Victoria, where it is listed as critically endangered on the Advisory List of Threatened Vertebrate Fauna [50]. Thus, the conservation status of *T. petersi* should be reassessed at both state and federal levels across its range.

With our study, plus two previous assessments [19,29], it is apparent that virtually all species of *Tympanocryptis* in southeastern Australia and southeastern Queensland are of conservation concern. These species of earless dragons, compared with other *Tympanocryptis*, occur in areas of eastern Australia that have been most impacted by habitat loss, fragmentation and degradation through urban expansion, agriculture and grazing [51]. We suggest that these dragons are particularly vulnerable to such impacts, resulting in population declines and loss.

The precipitous declines in these eastern Australian *Tympanocryptis* species should serve as an example from which lessons can be learnt for future habitat removal or modification in other regions of Australia. Such situations may occur again with, for example, pushes to expand and intensify agriculture and grazing in the tropical savannah grasslands [52]. Already, dramatic declines have been documented in the mammal fauna of the Northern Territory, reflecting patterns seen in southern and eastern Australia [53], which have been attributed to a complex interaction of factors including habitat loss, feral predators, invasive species and disease. It is of critical importance not only to accurately assess the conservation status of newly described and revised species of *Tympanocryptis* but to also implement appropriate management strategies, particularly taking into account the growing evidence that these species are extremely vulnerable to threatening processes.

Ethics. All samples used in this study were tissue samples held in museum collections.

Data accessibility. Additional text and data can be found in the electronic supplementary material. mtDNA sequences are available from Genbank, with accession numbers provided in electronic supplementary material, table S1. The mtDNA alignment, micro-CT cranial mesh files (PLY format) and landmark files are available in the Figshare digital repository and can be accessed at: https://doi.org/10.26188/5dbf56752bf10.

Authors' contributions. All authors contributed to analysis and interpretation of data and drafting of the manuscript. J.M., M.H., J.S. and S.D.S. contributed to study design and concept. K.C. collected external morphology data and CT scans; K.C. and J.S. undertook genetic analyses; K.C. and C.A.H. undertook morphological and geometric morphometric analyses. J.M., C.A.H., M.H. and K.C. wrote the paper.

Competing interests. We declare we have no competing interests.

Funding. Research support for this project was provided by the Department of Environment and Planning, ACT Government. C.H. is supported by an ARC DECRA (grant no. DE180100629).

Acknowledgements. We thank W. Osborne, T. McGrath and P. Robertson for their expertise and advice. Thank you to J. Black for technical support with micro X-ray CT scanning at the School of Earth Sciences, University of Melbourne. S.D.S. acknowledges the support of the Australian Research Council LP110200029.

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
