## [Reviewer comments · Royal Society Open Science]

Review History

RSOS-191166.R0 (Original submission)

Review form: Reviewer 1

Is the manuscript scientifically sound in its present form?

Yes

Are the interpretations and conclusions justified by the results?

Yes

Is the language acceptable?

Yes

Do you have any ethical concerns with this paper?

No

Have you any concerns about statistical analyses in this paper?

Yes

Recommendation?

Accept with minor revision (please list in comments)

Comments to the Author(s)

This manuscript, entitled "Integrating phylogeography and high-resolution x-ray CT reveals five new cryptic species and multiple hybrid zones among Australian earless dragons", aims to unravel the taxonomic structure of the Australian earless dragon species complex *Tympanocryptic lineata*. The authors used meristic characters, external linear measurements, 3D micro-CT data of the skull, and both mtDNA and nDNA data, to assign each specimen into twelve pre-defined taxa (seven previously described and five undescribed species of the *T. lineata* species complex). The authors show that 3D geometric morphometrics is a powerful tool to discover new cryptic species, and how that is crucial for informing conservation decisions. Specifically, they describe five new species and provide strong support for maintaining the taxonomic classification of seven already-described species.

Overall, I quite enjoyed reading this manuscript. It is well written, the authors have performed a lot of sound analyses, and it tackles a very important and exciting question on cryptic species. Therefore, I think this will be a very interesting contribution and I recommend it to be published in Royal Society Open.

However, I have a few minor concerns and questions regarding the overlap of data used in this study and Melville et al. (2019), as well as some of the analyses. I think it would be useful to clarify this to avoid confusion on which datasets were used. I also think the authors should expand further on the methodology so that it matches the Results section more closely, and makes it clearer to the reader. Similarly, it would be helpful if the authors justified some of the analyses chosen here. I have provided a few comments that I hope will help to improve this paper. However, some of the comments I am giving are a personal preference, so while I hope my comments will help to improve the paper, I do not expect the authors to perform or reply to all of them.

ABSTRACT:

Lines 27 & 36 - The authors mention twice that they use an evolutionary comparative approach, but most of the linear measurements and 3D geometric morphometrics analyses are not performed in a phylogenetic context (except estimating the strength of phylogenetic signal on cranial shape data). The authors should decide whether they would like to use phylogenetic comparative methods or change their text accordingly throughout the manuscript. If so, they should perform phylogenetic Principal Component Analyses (phyloPCA) instead of PCAs, phyloANOVAs instead of ANOVAs, etc. Otherwise, I do not think it is appropriate to claim that they used an evolutionary comparative approach. If the authors decided to use phylogenetic comparative methods (PCM), I think it would be good to mention limitations of PCM on such a small sample size (twelve species).

INTRODUCTION:

It is very well written.

METHODS:

L 116 - Even though all the specimens' information is provided in Supplementary Table S1, it would be good to provide sample sizes here. As well as that, I think the paper would be a bit clearer if the overlap between this manuscript and Melville et al. (2019) was explained further.

L 133-134 – Provide them in Supp Tables.

L 140-141 – Since the authors use geometric morphometrics (GM) to assess the skull shape, it would be worth using the log-shape ratios method (Mosimann, 1970), so that this size correction is comparable to the size correction in the Generalised Procrustes Analyses. That way both shape datasets would be comparable. See:

Mosimann, J. E. Size allometry: size and shape variables with characterizations of the lognormal and generalized gamma distributions. *J. Am. Stat. Assoc.* 65, 930–945 (1970).

Sherratt, E., Vidal-García, M., Anstis, M. Keogh, J.S. Adult frogs and tadpoles have different macroevolutionary patterns across the Australian continent. *Nature Ecol. Evol.* 1, 1385–1391 (2017)

L 150-151 – This sentence needs more information. Which variables were used in the discriminant function analysis? The authors use DFA, which is a parametric classification approach, but it might be worth considering a non-parametric method (e.g. Random Forests), depending on the data structure. It would be good for the authors to justify their choice either way.

L 158-160 – Were the landmarks placed on the whole cranium or just on the left side? It might be worth only looking at the left side of the cranium in order to keep the methodology consistent between linear measurements of the body and GM of the skull.

L 162 – damaged how? Perhaps it is worth not including specimen SMAR58187?

L 192-193 – This sentence needs a bit more detail. Was the strength of phylogenetic signal estimated in the Procrustes shape variables? It would be good to also assess the strength of phylogenetic signal in the other three sets of variables: (1) size-corrected body shape variables, (2) body size, and (3) centroid size of the head.

L 194 – What was the sampling? Please provide details on specimens and species.

L 195 – mention which mitochondrial genes

RESULTS:

L 217-220 – This needs to be explained in the Methods section

L 247-256 – See my previous comments on DFA (Methods section)

L 257-262 – See my comment on non-parametric classification methods. I really think it would be worth choosing a non-parametric method. As well as that, the sample size of each group should be higher (ideally) than the number of predictor variables when using DFA. The authors should provide this information in the Methods section.

L 283-294 – Please provide more details in the Methods section so it matches with this paragraph in Results.

L 312-313 – That is a strong statement that is not fully supported by the results. I assumed that the pairwise species comparisons of least squares were performed on Procrustes shape data, but the first two PCs just show 36% of the total shape variance.

L 324 – Can you also provide the results for mitochondrial genes?

L 332-333 – Please provide bootstrap values for the monophyletic support of *T. houstoni*. Also see my comment for L 194. If all species were included provide results here.

L 358 – 367 – This needs to be explained further in the Methods section.

L 389 – 1051 – These five species descriptions are important but also very long. I wonder whether they would be better suited in the Supplementary Materials.

DISCUSSION

I liked the discussion, especially the section on conservation implications. However, I generally feel like it would be good if the authors expanded further on what their results mean and put them and their methodological approach in context with the literature. I would personally be quite interested to see a further discussion on hybridization and incomplete lineage sorting in a biogeographic context.

L 1063-1064 – This is a very strong statement given the bootstrap support values for the monophyly of species A, and C. I think the authors should provide more information.

FIGURES

It would be good to see schematics of the linear measurements and the placement of the landmarks.

Fig. 2 – Which meristic and linear measurements?

Fig. 3 – See my previous comment

Review form: Reviewer 2 (Paul Doughty)

Is the manuscript scientifically sound in its present form?

No

Are the interpretations and conclusions justified by the results?

No

Is the language acceptable?

Yes

Do you have any ethical concerns with this paper?

No

Have you any concerns about statistical analyses in this paper?

Yes

Recommendation?

Major revision is needed (please make suggestions in comments)

Comments to the Author(s)

This paper analyses a complex of similar-looking dragon lizards from central Australia. These kinds of lizards are highly cryptic in their environments and also similar to each other phenotypically. The authors employ mt and nDNA to estimate divergences, and carry out external and internal (cranial) morphological measurements on the lineages deduced from the genetics.

I suspect these are all probably valid species, and the authors have done a good job in generating the various lines of evidence. The data sets are fairly clear to see, but I am nervous about some of the conflicts among data sets, and a lack of a wider perspective decision section that made sense of everything.

Part of the problem I believe is that this is a truly complex 'complex'. As such, the presentation of the data should not be business as usual, but go far beyond the normal so the reader can make sense of the patterns. The authors have done this a bit, but I was frequently confused trying to keep track track of everything while going through the manuscript many times.

To provide evidence that multiple species exist, the results of the mtDNA are used to group specimens, then external and internal morphology were used to look for separation. For the external morphology, there was some separation evident, especially for lineages C and E (Figs. 2 and 3). In the third paragraph of this section (starts with line 267) the percentage of correct scores are given, but I'd rather see tallies (e.g. not 75% but rather 6/8 for *houstoni*), as reporting the percentages gives an appearance that these are based on large sample sizes (they aren't). Looking at the separation of A-B-D groups, they appear to highly overlap.

For the internal cranial morphology, there seemed to be some evidence of separation of the groups (Fig. 5), but again things get crowded with the A-D-*houstoni* focal lineages here (although C+D seemed to drift away from A+B in the figure). For Table 2, I would like to see the significance of the pairwise tests after some kind of correcting for the sheer number of tests (e.g. Bonferroni correction). Many of the significant tests involved pairs of species that were more distantly related, and not the more contentious A-D groups (E is the exception, clearly standing out).

I was somewhat disappointed to see that the cranial morphology was used to test whether the mtDNA clades were species (fine), but abandoned entirely in the description section. If there were differences in cranial morphology, then there would surely be some useful diagnostic characters in the skull. After all, the problem with cryptic species is a lack of obvious diagnostic characters, so the potential of reporting good characters here has gone unrealised.

In the genetic data, there seem to be a very high proportion of hybrids. In addition, *rag1* was variably supportive of the species distinctiveness as well. The morphology was a bit hard to interpret in the text and images, but just looking at Species C it seemed different, and Species E is in a different lineage (called various things) and is distinct.

For the other species, it seemed they were species if you looked at them through this or that data. So, if Species C resembled A, B and D, then that could justify lumping C+D. But because C looks different from A, that provides evidence to counter the *rag1* pattern. If this sounds confusing, that's because the patterns and data really are confusing in this group!

Focus on:

3.5 Basis of taxonomic decisions

This section has two paragraphs: one on ITAX, and one on hybrids.

Application of the forthcoming ITAX method is introduced in section 3.5. It reads as though two lines of evidence are sufficient for calling something a species, and this is graphically represented in Fig. 7. Just as ITAX is a bit mysterious to me, so this figure based on the initial mtDNA findings was also hard to interpret. I would have preferred a different-looking bar across each node for an independent data set. So for B - it's *rag1* and mtDNA that are the two lines of evidence (because morph was not different from what I can see in the figures). Whereas for E - lots of lines of

evidence, lots of bars that indicate what the evidence was. Maybe reading the in press paper is necessary to grasp this method.

Hybrids. There seem to be quite a lot given the small sample sizes. A 'hybrid' presupposes you have two things. Another view of these data is that gene flow is still ongoing or recently stopped. This paragraph seems deal with each of these individuals as a way to assure the reader they're exceptions. I'd like to see the other point of view: that they're one big species, but then there's these funny differences that can't be ignored that lead the authors to conclude that, yes, these are good species.

Missing paragraph. For this 'taxonomic decision' section, I would have liked to see an extended discussion of the consequences of this or that decision. Yes, these things are difficult, especially with recently separated similar-looking entities. Still, I would feel more comfortable with any final decision if the authors ran through a few different scenarios of how to divide up the taxonomy. For instance, consider lumping A+B - why would this be a bad decision, and why would it be a good decision? The other three new species seem to get a pass on various combinations of evidence, and these could be made explicit too. Finally, the reader would want to know what is special about these dragons that they have these conflicting patterns (short limbs, philopatric, habitat specialists clinging to patches of clay/sand, etc.). Such a section would be interesting and justify more clearly the species decisions made.

Taxonomy. I have not done a detailed check of this section, but they stick to the main diagnostic characters and refer to Table 4 which is helpful. The images in life could be broken into multiple images to enlarge the lizards, rather than the thumbnail-like sizes they are now.

Minor issues:

- Terminology of groups: These seemed to vary inconsistently. Perhaps a single table or figure could be the terminological throw-down that all the text slavishly followed.
- Terminal tip labels: Always use museum voucher specimen numbers where available, not genetic code names.
- Grammar and typos: Manuscript should be combed through to clean many of these problems up. There were a heap.
- Reference formatting: A smorgasbord of styles both in the text and in the reference list. Stick to the journal format without deviation.

In conclusion, I think the manuscript has most of the essential information, but could do with some improvement from a big picture perspective at how the taxonomic decisions were made. The authors should be commended for attempting to incorporate so many different lines of evidence, but alas it was the conflict among the data sets that was confusing to me.

Paul Doughty
August 2019
WA Museum

Decision letter (RSOS-191166.R0)

16-Aug-2019

Dear Dr Melville,

The editors assigned to your paper ("Integrating phylogeography and high-resolution x-ray CT reveals five new cryptic species and multiple hybrid zones among Australian earless dragons") have now received comments from reviewers. We would like you to revise your paper in accordance with the referee and Associate Editor suggestions which can be found below (not including confidential reports to the Editor). Please note this decision does not guarantee eventual acceptance.

Please submit a copy of your revised paper before 08-Sep-2019. Please note that the revision deadline will expire at 00.00am on this date. If we do not hear from you within this time then it will be assumed that the paper has been withdrawn. In exceptional circumstances, extensions may be possible if agreed with the Editorial Office in advance. We do not allow multiple rounds of revision so we urge you to make every effort to fully address all of the comments at this stage. If deemed necessary by the Editors, your manuscript will be sent back to one or more of the original reviewers for assessment. If the original reviewers are not available, we may invite new reviewers.

- Data accessibility

If you wish to submit your supporting data or code to Dryad (<http://datadryad.org/>), or modify your current submission to dryad, please use the following link:
<http://datadryad.org/submit?journalID=RSOS&manu=RSOS-191166>

- **Competing interests**

- **Authors' contributions**

- **Acknowledgements**

- **Funding statement**

Kind regards,

Andrew Dunn

on behalf of Dr Julia Brenda Desojo (Associate Editor) and Kevin Padian (Subject Editor)
openscience@royalsociety.org

Editor comments (Prof Kevin Padian)

Thanks for your submission. As you will see both reviewers had some substantial concerns but these do not seem to be fatal; however they will have to be addressed assiduously, so please provide detailed responses with your revised manuscript. If you need more time please contact the editorial office. Best wishes.

Comments to Author:

Reviewers' Comments to Author:

Reviewer: 1

Comments to the Author(s)

This manuscript, entitled "Integrating phylogeography and high-resolution x-ray CT reveals five new cryptic species and multiple hybrid zones among Australian earless dragons", aims to unravel the taxonomic structure of the Australian earless dragon species complex *Tympanocryptus lineata*. The authors used meristic characters, external linear measurements, 3D micro-CT data of the skull, and both mtDNA and nDNA data, to assign each specimen into twelve pre-defined taxa (seven previously described and five undescribed species of the *T. lineata* species complex). The authors show that 3D geometric morphometrics is a powerful tool to discover new cryptic species, and how that is crucial for informing conservation decisions. Specifically, they describe five new species and provide strong support for maintaining the taxonomic classification of seven already-described species.

Overall, I quite enjoyed reading this manuscript. It is well written, the authors have performed a lot of sound analyses, and it tackles a very important and exciting question on cryptic species. Therefore, I think this will be a very interesting contribution and I recommend it to be published in Royal Society Open.

However, I have a few minor concerns and questions regarding the overlap of data used in this study and Melville et al. (2019), as well as some of the analyses. I think it would be useful to clarify this to avoid confusion on which datasets were used. I also think the authors should expand further on the methodology so that it matches the Results section more closely, and makes it clearer to the reader. Similarly, it would be helpful if the authors justified some of the analyses chosen here. I have provided a few comments that I hope will help to improve this paper. However, some of the comments I am giving are a personal preference, so while I hope my comments will help to improve the paper, I do not expect the authors to perform or reply to all of them.

ABSTRACT:

Lines 27 & 36 - The authors mention twice that they use an evolutionary comparative approach, but most of the linear measurements and 3D geometric morphometrics analyses are not performed in a phylogenetic context (except estimating the strength of phylogenetic signal on cranial shape data). The authors should decide whether they would like to use phylogenetic comparative methods or change their text accordingly throughout the manuscript. If so, they should perform phylogenetic Principal Component Analyses (phyloPCA) instead of PCAs, phyloANOVAs instead of ANOVAs, etc. Otherwise, I do not think it is appropriate to claim that they used an evolutionary comparative approach. If the authors decided to use phylogenetic comparative methods (PCM), I think it would be good to mention limitations of PCM on such a small sample size (twelve species).

INTRODUCTION:

It is very well written.

METHODS:

L 116 - Even though all the specimens' information is provided in Supplementary Table S1, it would be good to provide sample sizes here. As well as that, I think the paper would be a bit clearer if the overlap between this manuscript and Melville et al. (2019) was explained further.

L 133-134 – Provide them in Supp Tables.

L 140-141 – Since the authors use geometric morphometrics (GM) to assess the skull shape, it would be worth using the log-shape ratios method (Mosimann, 1970), so that this size correction is comparable to the size correction in the Generalised Procrustes Analyses. That way both shape datasets would be comparable. See:

Mosimann, J. E. Size allometry: size and shape variables with characterizations of the lognormal and generalized gamma distributions. *J. Am. Stat. Assoc.* 65, 930–945 (1970).

Sherratt, E., Vidal-García, M., Anstis, M. Keogh, J.S. Adult frogs and tadpoles have different macroevolutionary patterns across the Australian continent. *Nature Ecol. Evol.* 1, 1385–1391 (2017)

L 150-151 – This sentence needs more information. Which variables were used in the discriminant function analysis? The authors use DFA, which is a parametric classification approach, but it might be worth considering a non-parametric method (e.g. Random Forests), depending on the data structure. It would be good for the authors to justify their choice either way.

L 158-160 – Were the landmarks placed on the whole cranium or just on the left side? It might be worth only looking at the left side of the cranium in order to keep the methodology consistent between linear measurements of the body and GM of the skull.

L 162 – damaged how? Perhaps it is worth not including specimen SMAR58187?

L 192-193 – This sentence needs a bit more detail. Was the strength of phylogenetic signal estimated in the Procrustes shape variables? It would be good to also assess the strength of phylogenetic signal in the other three sets of variables: (1) size-corrected body shape variables, (2) body size, and (3) centroid size of the head.

L 194 – What was the sampling? Please provide details on specimens and species.

L 195 – mention which mitochondrial genes

RESULTS:

L 217-220 – This needs to be explained in the Methods section

L 247-256 – See my previous comments on DFA (Methods section)

L 257-262 – See my comment on non-parametric classification methods. I really think it would be worth choosing a non-parametric method. As well as that, the sample size of each group should be higher (ideally) than the number of predictor variables when using DFA. The authors should provide this information in the Methods section.

L 283-294 – Please provide more details in the Methods section so it matches with this paragraph in Results.

L 312-313 – That is a strong statement that is not fully supported by the results. I assumed that the pairwise species comparisons of least squares were performed on Procrustes shape data, but the first two PCs just show 36% of the total shape variance.

L 324 – Can you also provide the results for mitochondrial genes?

L 332-333 – Please provide bootstrap values for the monophyletic support of *T. houstoni*. Also see my comment for L 194. If all species were included provide results here.

L 358 – 367 – This needs to be explained further in the Methods section.

L 389 – 1051 – These five species descriptions are important but also very long. I wonder whether they would be better suited in the Supplementary Materials.

DISCUSSION

I liked the discussion, especially the section on conservation implications. However, I generally feel like it would be good if the authors expanded further on what their results mean and put them and their methodological approach in context with the literature. I would personally be quite interested to see a further discussion on hybridization and incomplete lineage sorting in a biogeographic context.

L 1063-1064 – This is a very strong statement given the bootstrap support values for the monophyly of species A, and C. I think the authors should provide more information.

FIGURES

It would be good to see schematics of the linear measurements and the placement of the landmarks.

Fig. 2 – Which meristic and linear measurements?

Fig. 3 – See my previous comment

Reviewer: 2

Comments to the Author(s)

This paper analyses a complex of similar-looking dragon lizards from central Australia. These kinds of lizards are highly cryptic in their environments and also similar to each other phenotypically. The authors employ mt and nDNA to estimate divergences, and carry out external and internal (cranial) morphological measurements on the lineages deduced from the genetics.

I suspect these are all probably valid species, and the authors have done a good job in generating the various lines of evidence. The data sets are fairly clear to see, but I am nervous about some of the conflicts among data sets, and a lack of a wider perspective decision section that made sense of everything.

Part of the problem I believe is that this is a truly complex 'complex'. As such, the presentation of the data should not be business as usual, but go far beyond the normal so the reader can make sense of the patterns. The authors have done this a bit, but I was frequently confused trying to keep track track of everything while going through the manuscript many times.

To provide evidence that multiple species exist, the results of the mtDNA are used to group specimens, then external and internal morphology were used to look for separation. For the external morphology, there was some separation evident, especially for lineages C and E (Figs. 2 and 3). In the third paragraph of this section (starts with line 267) the percentage of correct scores are given, but I'd rather see tallies (e.g. not 75% but rather 6/8 for *houstoni*), as reporting the

percentages gives an appearance that these are based on large sample sizes (they aren't). Looking at the separation of A-B-D groups, they appear to highly overlap.

For the internal cranial morphology, there seemed to be some evidence of separation of the groups (Fig. 5), but again things get crowded with the A-D-houstoni focal lineages here (although C+D seemed to drift away from A+B in the figure). For Table 2, I would like to see the significance of the pairwise tests after some kind of correcting for the sheer number of tests (e.g. Bonferroni correction). Many of the significant tests involved pairs of species that were more distantly related, and not the more contentious A-D groups (E is the exception, clearly standing out).

I was somewhat disappointed to see that the cranial morphology was used to test whether the mtDNA clades were species (fine), but abandoned entirely in the description section. If there were differences in cranial morphology, then there would surely be some useful diagnostic characters in the skull. After all, the problem with cryptic species is a lack of obvious diagnostic characters, so the potential of reporting good characters here has gone unrealised.

In the genetic data, there seem to be a very high proportion of hybrids. In addition, rag1 was variably supportive of the species distinctiveness as well. The morphology was a bit hard to interpret in the text and images, but just looking at Species C it seemed different, and Species E is in a different lineage (called various things) and is distinct.

For the other species, it seemed they were species if you looked at them through this or that data. So, if Species C resembled A, B and D, then that could justify lumping C+D. But because C looks different from A, that provides evidence to counter the rag1 pattern. If this sounds confusing, that's because the patterns and data really are confusing in this group!

Focus on:

3.5 Basis of taxonomic decisions

This section has two paragraphs: one on ITAX, and one on hybrids.

Application of the forthcoming ITAX method is introduced in section 3.5. It reads as though two lines of evidence are sufficient for calling something a species, and this is graphically represented in Fig. 7. Just as ITAX is a bit mysterious to me, so this figure based on the initial mtDNA findings was also hard to interpret. I would have preferred a different-looking bar across each node for an independent data set. So for B - it's rag1 and mtDNA that are the two lines of evidence (because morph was not different from what I can see in the figures). Whereas for E - lots of lines of evidence, lots of bars that indicate what the evidence was. Maybe reading the in press paper is necessary to grasp this method.

Hybrids. There seem to be quite a lot given the small sample sizes. A 'hybrid' presupposes you have two things. Another view of these data is that gene flow is still ongoing or recently stopped. This paragraph seems deal with each of these individuals as a way to assure the reader they're exceptions. I'd like to see the other point of view: that they're one big species, but then there's these funny differences that can't be ignored that lead the authors to conclude that, yes, these are good species.

Missing paragraph. For this 'taxonomic decision' section, I would have liked to see an extended discussion of the consequences of this or that decision. Yes, these things are difficult, especially with recently separated similar-looking entities. Still, I would feel more comfortable with any final decision if the authors ran through a few different scenarios of how to divide up the taxonomy. For instance, consider lumping A+B - why would this be a bad decision, and why would it be a good decision? The other three new species seem to get a pass on various

combinations of evidence, and these could be made explicit too. Finally, the reader would want to know what is special about these dragons that they have these conflicting patterns (short limbs, philopatric, habitat specialists clinging to patches of clay/sand, etc.). Such a section would be interesting and justify more clearly the species decisions made.

Taxonomy. I have not done a detailed check of this section, but they stick to the main diagnostic characters and refer to Table 4 which is helpful. The images in life could be broken into multiple images to enlarge the lizards, rather than the thumbnail-like sizes they are now.

Minor issues:

- Terminology of groups: These seemed to vary inconsistently. Perhaps a single table or figure could be the terminological throw-down that all the text slavishly followed.
- Terminal tip labels: Always use museum voucher specimen numbers where available, not genetic code names.
- Grammar and typos: Manuscript should be combed through to clean many of these problems up. There were a heap.
- Reference formatting: A smorgasbord of styles both in the text and in the reference list. Stick to the journal format without deviation.

In conclusion, I think the manuscript has most of the essential information, but could do with some improvement from a big picture perspective at how the taxonomic decisions were made. The authors should be commended for attempting to incorporate so many different lines of evidence, but alas it was the conflict among the data sets that was confusing to me.

Paul Doughty
August 2019
WA Museum

Author's Response to Decision Letter for (RSOS-191166.R0)

See Appendix A.

Decision letter (RSOS-191166.R1)

09-Oct-2019

Dear Dr Melville,

I am pleased to inform you that your manuscript entitled "Integrating phylogeography and high-resolution x-ray CT reveals five new cryptic species and multiple hybrid zones among Australian earless dragons" is now accepted for publication in Royal Society Open Science.

Please note that, at present, the GenBank accessions provided within Table S1 do not yet appear to be live; this process can take a few days (due to the GenBank validation checks), so we would be grateful if you could please let us know once these accessions are formally available. For help with this, please contact: help@ncbi.nlm.nih.gov

Once this process is complete, you can expect to receive a proof of your article in the near future. Please contact the editorial office (openscience_proofs@royalsociety.org and openscience@royalsociety.org) to let us know if you are likely to be away from e-mail contact -- if you are going to be away, please nominate a co-author (if available) to manage the proofing process, and ensure they are copied into your email to the journal.

With best regards,

on behalf of Dr Julia Brenda Desojo (Associate Editor) and Kevin Padian (Subject Editor)
openscience@royalsociety.org

Editor's Comments to Author:

Thanks for your attention to the concerns of the reviewers; we're prepared to accept this, and we wish you the best in your future work.

Follow Royal Society Publishing on Twitter: [@RSocPublishing](https://twitter.com/RSocPublishing)
Follow Royal Society Publishing on Facebook:
<https://www.facebook.com/RoyalSocietyPublishing.FanPage/>
Read Royal Society Publishing's blog: <https://blogs.royalsociety.org/publishing/>

Appendix A

Comments to Author:

Reviewers' Comments to Author:

Reviewer: 1

Comments to the Author(s)

This manuscript, entitled "Integrating phylogeography and high-resolution x-ray CT reveals five new cryptic species and multiple hybrid zones among Australian earless dragons", aims to unravel the taxonomic structure of the Australian earless dragon species complex *Tympanocryptic lineata*. The authors used meristic characters, external linear measurements, 3D micro-CT data of the skull, and both mtDNA and nDNA data, to assign each specimen into twelve pre-defined taxa (seven previously described and five undescribed species of the *T. lineata* species complex). The authors show that 3D geometric morphometrics is a powerful tool to discover new cryptic species, and how that is crucial for informing conservation decisions. Specifically, they describe five new species and provide strong support for maintaining the taxonomic classification of seven already-described species.

Overall, I quite enjoyed reading this manuscript. It is well written, the authors have performed a lot of sound analyses, and it tackles a very important and exciting question on cryptic species. Therefore, I think this will be a very interesting contribution and I recommend it to be published in Royal Society Open.

However, I have a few minor concerns and questions regarding the overlap of data used in this study and Melville et al. (2019), as well as some of the analyses. I think it would be useful to clarify this to avoid confusion on which datasets were used. I also think the authors should expand further on the methodology so that it matches the Results section more closely, and makes it clearer to the reader. Similarly, it would be helpful if the authors justified some of the analyses chosen here. I have provided a few comments that I hope will help to improve this paper. However, some of the comments I am giving are a personal preference, so while I hope my comments will help to improve the paper, I do not expect the authors to perform or reply to all of them.

We thank Reviewer 1's support of our manuscript. We note their minor concerns in regard to the data overlap with Melville et al., 2019. We have made revisions (detailed with each of the reviewers comments below) to strength the clarity of the distinction of data and analyses between the two papers. In addition, we agree with the reviewer that some additional details are needed in the methods to better explain sampling regimes and analytical approaches – we have now provided extensive revisions with more details around these issues (detailed) below. We believe, that in addressing the reviewers comments that our paper's readability and clarity has been improved.

ABSTRACT:

Lines 27 & 36 – The authors mention twice that they use an evolutionary comparative approach, but most of the linear measurements and 3D geometric morphometrics analyses are not performed in a phylogenetic context (except estimating the strength of phylogenetic signal on cranial shape data). The authors should decide whether they would like to use phylogenetic comparative methods or change their text accordingly throughout the manuscript. If so, they should perform phylogenetic Principal Component Analyses (phyloPCA) instead of PCAs, phyloANOVAs instead of ANOVAs, etc. Otherwise, I do not think it is appropriate to claim that they used an evolutionary comparative approach. If the authors decided to use phylogenetic comparative methods (PCM), I think it would be good to mention limitations of PCM on such a small sample size (twelve species).

We have changed the term “evolutionary comparative” to “integrative” in the abstract.

INTRODUCTION:

It is very well written.

METHODS:

L 116 – Even though all the specimens’ information is provided in Supplementary Table S1, it would be good to provide sample sizes here. As well as that, I think the paper would be a bit clearer if the overlap between this manuscript and Melville et al. (2019) was explained further.

We have provided details of sample sizes based on initial morphological assessments.

L 133-134 – Provide them in Supp Tables.

We have provided the details of the characters used in the analysis in the text.

L 140-141 – Since the authors use geometric morphometrics (GM) to assess the skull shape, it would be worth using the log-shape ratios method (Mosimann, 1970), so that this size correction is comparable to the size correction in the Generalised Procrustes Analyses. That way both shape datasets would be comparable. See:

Mosimann, J. E. Size allometry: size and shape variables with characterizations of the lognormal and generalized gamma distributions. J. Am. Stat. Assoc. 65, 930–945 (1970).

Sherratt, E., Vidal-García, M., Anstis, M. Keogh, J.S. Adult frogs and tadpoles have different macroevolutionary patterns across the Australian continent. Nature Ecol. Evol. 1, 1385–1391 (2017)

Although we agree that the log-shape ratios method would remove allometric influence when looking at shape, in the analysis here (PCA) we are not looking at shape difference, instead we are looking at a combination of meristic (count) and simple linear measures. We are not undertaking a geometric morphometric shape analysis in this section, rather looking at which variable best distinguish species. Hence the use of a Discriminant function analysis (DFA). For this reason we believe that a linear method for removing the effect of size from measures is appropriate.

L 150-151 – This sentence needs more information. Which variables were used in the discriminant function analysis? The authors use DFA, which is a parametric classification approach, but it might be worth considering a non-parametric method (e.g. Random Forests), depending on the data structure. It would be good for the authors to justify their choice either way.

We believe that a DFA is an appropriate test for the small number of variables being used in this case, with no significant assumptions being violated. Samples sizes are adequate, variables used are appropriate. However, we do acknowledge that information provided in the methods were not detailed enough for readers to determine this. We have now included a statement, as requested as to why this test was used and details about pre-analysis checks that were used. “Prior to analyses univariate tests of each morphological variable were used to identify outliers and ensure data did not differ significantly from a normal distribution.”

L 158-160 – Were the landmarks placed on the whole cranium or just on the left side? It might be worth only looking at the left side of the cranium in order to keep the methodology consistent between linear measurements of the body and GM of the skull.

These measurements, as detailed in the methods were “placed across the surface of the cranium”, and were not on one side of the head. It is an interesting suggestion to put them just to the left side of the head, however, to estimate the shape of the cranium these points and shape measures span the left and right side of the cranium. Thus, it is not logical to limit the quantification of shape to a single side. In addition, we don't think that limiting our analysis of cranial shape so that it matches the external measures is necessary, as both the cranial shape and external morphological analyses are stand-alone analyses.

L 162 – damaged how? Perhaps it is worth not including specimen SMAR58187?

These are museum specimens and in some instances that have had some small mechanical damage that can only be seen once a scan is completed. Our sample size is such that we made the decision that these three scans needed to be retained to ensure the statistical power of our analyses.

L 192-193 – This sentence needs a bit more detail. Was the strength of phylogenetic signal estimated in the Procrustes shape variables? It would be good to also assess the strength of phylogenetic signal in the other three sets of variables: (1) size-corrected body shape variables, (2) body size, and (3) centroid size of the head.

As detailed above, the analyses of external morphology was not an analysis of shape. We used these analyses, as outlined, to separate species based on commonly measured characters for taxonomy – not on shape. These simple characters are readily interpretable for field-based or taxonomic identification of animals. For these reasons, an analysis of phylogenetic signal for shape in these characters is not needed and would overly complicate an already complex study. Additional text was added to the phylogenetic signal sentence that the test is “based on the above phylogeny and the Procrustes shape coordinates, as well as individual centroid size”. The results of the latter were reported in the results.

L 194 - What was the sampling? Please provide details on specimens and species.

A reference to Supplementary Table S1 has been added. This table indicates which specimens were scanned.

L 195 – mention which mitochondrial genes

Details have been added.

RESULTS:

L 217-220 – This needs to be explained in the Methods section

Details have been added to the methods.

L 247-256 – See my previous comments on DFA (Methods section)

We address this comment above in the methods section.

L 257-262 – See my comment on non-parametric classification methods. I really think it would be worth choosing a non-parametric method. As well as that, the sample size of each group should be higher (ideally) than the number of predictor variables when using DFA. The authors should provide this information in the Methods section.

As detailed in our comment about this in the methods section. We believe that a DFA is an appropriate test for the small number of variables being used in this case, with no significant assumptions being violated. Samples sizes are adequate, variables used are appropriate. However, we do acknowledge that information provided in the methods were not detailed enough for readers to determine this. We have now included a statement, as requested as to why this test was used and details about pre-analysis checks that were used: “Prior to analyses univariate tests of each morphological variable were used to identify outliers and ensure data did not differ significantly from a normal distribution.”

L 283-294 – Please provide more details in the Methods section so it matches with this paragraph in Results.

We apologise – we did have a paragraph about this in the methods section, which appears to have been inadvertently dropped through edits. We have put this sentence back into the methods section: “In addition to these meristic and metric measures, all specimens were qualitatively assessed for variation in scales, patterns and colour that may prove useful to distinguish species.”

L 312-313 – That is a strong statement that is not fully supported by the results. I assumed that the pairwise species comparisons of least squares were performed on Procrustes shape data, but the first two PCs just show 36% of the total shape variance.

We do discuss the results from the different data sources in the taxonomic decisions section, so we have added this detail to the sentence: “Taxonomic groupings in morphospace appear to reflect the above pairwise results (detailed further in taxonomic decisions section below), with the first two PC axes explaining 36% of the total shape variance (Fig. 5).”

L 324 – Can you also provide the results for mitochondrial genes?

We did not provided these mtDNA details in this paper, as it has already been published in the Melville et al. 2019 paper. We just use the dataset to present a full phylogenetic tree. It is not usual to re-report these stats and we don't want to confuse readers. However, the nuclear data is unpublished and this is the first time it is reported.

L 332-333 – Please provide bootstrap values for the monophyletic support of *T. houstoni*. Also see my comment for L 194. If all species were included provide results here.

We cannot provide this support value as *T. houstoni* was not supported as monophyletic – the text clearly states: “Of the already named species, *T. houstoni* was **not** supported as a monophyletic lineage, probably a result of incomplete lineage sorting in the nuclear gene.”

In the paragraph following this we detail the phylogenetic relationships between all lineages: “The phylogenetic relationships between lineages differed between the mtDNA and nuclear phylogeny. Unlike the mtDNA phylogeny, *T. l. macra* was resolved as a member of the Pebble Dragon clade in the RAG1 phylogeny, along with the *T. cephalus* clade, *T. intima* and Species E. The remaining lineages of *Tympanocryptis* formed a second lineage, within which were two sister lineages: (1) the *T. tetraporphora* clade; and (2) the remaining *T. lineata*-group samples. This second clade of *T. lineata*-group samples comprised of a single basal *T.*

houstoni sample and the two sister lineages: (1) other *T. houstoni* samples, the GED clade and Species D; and (2) Species A, B and C.”

We have added an additional detail in the methods – a reference to Supplementary Table S1 as this table indicates which specimens were sequenced for which genes.

L 358 – 367 – This needs to be explained further in the Methods section.

We outlined in this paragraph what the ITAX methods is however, based on this requests we have removed this sentence from the results to avoid repetition and added a short section into the methods. We have provided some additional details of the method, as requested.

L 389 – 1051 – These five species descriptions are important but also very long. I wonder whether they would be better suited in the Supplementary Materials.

This paper is primarily a taxonomic paper outlining how integrative methods can be used in cryptic lineages, thus a key part of this paper is the taxonomic section. We believe that it is important to leave the taxonomy section in the main body of the paper. We have reduced the length by putting a number of the image plates into the supplementary materials.

DISCUSSION

I liked the discussion, especially the section on conservation implications. However, I generally feel like it would be good if the authors expanded further on what their results mean and put them and their methodological approach in context with the literature. I would personally be quite interested to see a further discussion on hybridization and incomplete lineage sorting in a biogeographic context.

Thank you for the support of the discussion and although we like the idea of expanding the details on hybridization and incomplete lineage sorting we feel that we are wary of expanding further, as we believe additional sampling and additional genetic data (more genes or genomics) are needed to explore this topic in more details. Our data provides a strong basis to delimit these cryptic species but we do not want to overstep these data. We believe such details are beyond the scope of our study. However, our study provides a strong foundation for future research. To highlight this point we have added a sentence to the end of the 1st paragraph in the discussion: “Our study provides foundational evidence of hybridization and incomplete lineage sorting, where additional sampling and genetic data (e.g., genomics) are needed to explore this topic in more detail.”

L 1063-1064 – This is a very strong statement given the bootstrap support values for the monophyly of species A, and C. I think the authors should provide more information.

We have now modified this sentence to read: “More generally, our quantification of morphological variation as part of whole-evidence taxonomy offered greater confidence in the existence of separate species across these cryptic lineages.”

FIGURES

It would be good to see schematics of the linear measurements and the placement of the landmarks.

The linear measurements used in this study are very standard for taxonomic studies and we believe that such a figure is not needed for these measures. However, we have now included and figure of the landmarks used in the Figure supplementary materials.

Fig. 2 – Which meristic and linear measurements?

This information has been added into the figure caption.

Fig. 3 – See my previous comment

This information has been added into the figure caption.

Reviewer: 2

Comments to the Author(s)

This paper analyses a complex of similar-looking dragon lizards from central Australia. These kinds of lizards are highly cryptic in their environments and also similar to each other phenotypically. The authors employ mt and nDNA to estimate divergences, and carry out external and internal (cranial) morphological measurements on the lineages deduced from the genetics.

I suspect these are all probably valid species, and the authors have done a good job in generating the various lines of evidence. The data sets are fairly clear to see, but I am nervous about some of the conflicts among data sets, and a lack of a wider perspective decision section that made sense of everything.

Part of the problem I believe is that this is a truly complex 'complex'. As such, the presentation of the data should not be business as usual, but go far beyond the normal so the reader can make sense of the patterns. The authors have done this a bit, but I was frequently confused trying to keep track of everything while going through the manuscript many times.

To provide evidence that multiple species exist, the results of the mtDNA are used to group specimens, then external and internal morphology were used to look for separation. For the external morphology, there was some separation evident, especially for lineages C and E (Figs. 2 and 3). In the third paragraph of this section (starts with line 267) the percentage of correct scores are given, but I'd rather see tallies (e.g. not 75% but rather 6/8 for houstoni), as reporting the percentages gives an appearance that these are based on large sample sizes (they aren't). Looking at the separation of A-B-D groups, they appear to highly overlap.

We have updated the DFA results section to include both the percentages and also the counts, as suggested by Reviewer 2.

For the internal cranial morphology, there seemed to be some evidence of separation of the groups (Fig. 5), but again things get crowded with the A-D-houstoni focal lineages here (although C+D seemed to drift away from A+B in the figure). For Table 2, I would like to see the significance of the pairwise tests after some kind of correcting for the sheer number of tests (e.g. Bonferroni correction). Many of the significant tests involved pairs of species that were more distantly related, and not the more contentious A-D groups (E is the exception, clearly standing out).

The results of the pairwise tests as implemented in geomorph represent the actual observed probability of the pairwise comparison relative to the distribution of outcomes from permutation. Since the random residual permutation procedure (RRPP) used is designed to handle high dimensional data and large variable-to-specimen ratios, and has been validated as having

acceptable type I error rates (see Adams et al 2018), Bonferroni or other such corrections are not commonly used.

Adams, D. C., & Collyer, M. L. (2018). Multivariate phylogenetic comparative methods: Evaluations, comparisons, and recommendations. *Systematic Biology*, 67, 14–31.

I was somewhat disappointed to see that the cranial morphology was used to test whether the mtDNA clades were species (fine), but abandoned entirely in the description section. If there were differences in cranial morphology, then there would surely be some useful diagnostic characters in the skull. After all, the problem with cryptic species is a lack of obvious diagnostic characters, so the potential of reporting good characters here has gone unrealised.

The geometric morphometric shape analysis was used to delimit species, however, rather than to identify a lizard in hand, which is what the taxonomic section we aim to achieve. However, we can understand the reviewers point of view. In line with this we have added a sentence to each species description section describing the general shape of the skull. For example: “Head shape, very wide skull with very short snout.”

In the genetic data, there seem to be a very high proportion of hybrids. In addition, rag1 was variably supportive of the species distinctiveness as well. The morphology was a bit hard to interpret in the text and images, but just looking at Species C it seemed different, and Species E is in a different lineage (called various things) and is distinct.

We address these comments below under the more specific points raised.

For the other species, it seemed they were species if you looked at them through this or that data. So, if Species C resembled A, B and D, then that could justify lumping C+D. But because C looks different from A, that provides evidence to counter the rag1 pattern. If this sounds confusing, that's because the patterns and data really are confusing in this group!

Focus on:

3.5 Basis of taxonomic decisions

This section has two paragraphs: one on ITAX, and one on hybrids.

We have added in further details to the ITAX results summarising the lines of evidence supporting lineages, further supported by a new table (Table 3).

Application of the forthcoming ITAX method is introduced in section 3.5. It reads as though two lines of evidence are sufficient for calling something a species, and this is graphically represented in Fig. 7. Just as ITAX is a bit mysterious to me, so this figure based on the initial mtDNA findings was also hard to interpret. I would have preferred a different-looking bar across each node for an independent data set. So for B - it's rag1 and mtDNA that are the two lines of evidence (because morph was not different from what I can see in the figures). Whereas for E - lots of lines of evidence, lots of bars that indicate what the evidence was. Maybe reading the in press paper is necessary to grasp this method.

We can understand that this is confusing. And although it is an interesting suggestion to add the lines of evidence to the mtDNA ITAX tree, this is not possible as the morphology delimitation is based on a pairwise analysis between lineages. For this reason to aid in the ease of interpretation we

have added in an additional table (Table 3) that summarises the morphological lines of evidence between the lineages.

Hybrids. There seem to be quite a lot given the small sample sizes. A 'hybrid' presupposes you have two things. Another view of these data is that gene flow is still ongoing or recently stopped. This paragraph seems deal with each of these individuals as a way to assure the reader they're exceptions. I'd like to see the other point of view: that they're one big species, but then there's these funny differences that can't be ignored that lead the authors to conclude that, yes, these are good species.

Although it may appear that we have a high level of hybrids – this is an artificial inflation as we focussed sampling in the region of the Gawler ranges and Eyre Peninsula in which this hybridization appears to be focussed. Thus, the level of hybridization cannot be interpreted from these results – and as explained in the results there was no evidence (genetic or morphological) of hybridization outside this contact area.

Missing paragraph. For this 'taxonomic decision' section, I would have liked to see an extended discussion of the consequences of this or that decision. Yes, these things are difficult, especially with recently separated similar-looking entities. Still, I would feel more comfortable with any final decision if the authors ran through a few different scenarios of how to divide up the taxonomy. For instance, consider lumping A+B - why would this be a bad decision, and why would it be a good decision? The other three new species seem to get a pass on various combinations of evidence, and these could be made explicit too. Finally, the reader would want to know what is special about these dragons that they have these conflicting patterns (short limbs, philopatric, habitat specialists clinging to patches of clay/sand, etc.). Such a section would be interesting and justify more clearly the species decisions made.

We have added further details to the taxonomic decisions section detailing the evidence on which decisions were made, including the additional table summarising this. We also detail habitat requirements of species within the taxonomy section. These details are not extensive as currently little is known about these taxa or their habits. Most of this species occur in extremely remote regions of Australia and further research, beyond the scope of this study, needs to be undertaken to better understand the relationship between morphology and habitat.

Taxonomy. I have not done a detailed check of this section, but they stick to the main diagnostic characters and refer to Table 4 which is helpful. The images in life could be broken into multiple images to enlarge the lizards, rather than the thumbnail-like sizes they are now.

Minor issues:

- **Terminology of groups:** These seemed to vary inconsistently. Perhaps a single table or figure could be the terminological throw-down that all the text slavishly followed.
- **Terminal tip labels:** Always use museum voucher specimen numbers where available, not genetic code names.
- **Grammar and typos:** Manuscript should be combed through to clean many of these problems up. There were a heap.
- **Reference formatting:** A smorgasbord of styles both in the text and in the reference list. Stick to the journal format without deviation.

We have gone through the manuscript to address these comments. In terms of the Terminal Tip labels – we used the genetic code in some instances, as they were previously published genetic

sequence, which would include these details in their publication – in this way readers can distinguish between our data and previously published data.

In conclusion, I think the manuscript has most of the essential information, but could do with some improvement from a big picture perspective at how the taxonomic decisions were made. The authors should be commended for attempting to incorporate so many different lines of evidence, but alas it was the conflict among the data sets that was confusing to me.

**Paul Doughty
August 2019
WA Museum**